



# Cross-evaluating WRF-Chem v4.1.2, TROPOMI, APEX and in situ NO₂ measurements over Antwerp, Belgium

Catalina Poraicu[1], Jean-François Müller[1], Trissevgeni Stavrakou[1], Dominique Fonteyn[1], Frederik Tack[1], Felix Deutsch[2], Quentin Laffineur[3], Roeland Van Malderen[3], and Nele Veldeman[2]

[1]Royal Belgian Institute for Space Aeronomy (BIRA-IASB), Ringlaan 3, 1180 Brussels, Belgium
[2]Flemish Institute for Technological Research (VITO), Boeretang 200, 2400 Mol, Belgium
[3]Royal Meteorological Institute of Belgium (RMI), Ringlaan 3, 1180 Brussels, Belgium

*Correspondence to*: Catalina Poraicu (catalina.poraicu@aeronomie.be)

**Abstract.** The Weather Research and Forecasting model coupled with Chemistry (WRF-Chem) is employed as an intercomparison tool for validating Tropospheric Monitoring Instrument (TROPOMI) satellite NO₂ retrievals against high-resolution Airborne Prism EXperiment (APEX) remote sensing observations performed in June 2019 in the region of Antwerp, a major hotspot of NO₂ pollution in Europe. The model is first evaluated using meteorological and chemical observations in this area. Sensitivity simulations varying the model planetary layer boundary (PBL) parameterization were conducted for a 3-day period in June 2019, indicating a general good performance of most parameterizations against meteorological data (namely ceilometer, surface meteorology and balloon measurements), except for a moderate overestimation (~1 m s$^{-1}$) of near-surface wind speed. On average, all but one PBL schemes reproduce fairly well the surface NO₂ measurements at stations of the Belgian Interregional Environmental Agency, although surface NO₂ is generally underestimated during the day (between -4.3 and -25.1% on average) and overestimated at night (8.2-77.3%). This discrepancy in the diurnal evolution arises despite (1) implementing a detailed representation of the diurnal cycle of emissions (Crippa et al., 2020), and (2) correcting the modelled concentrations to account for measurement interferences due to NO$_y$ reservoir species, which increases NO₂ concentrations by about 20% during the day. The model is further evaluated by comparing a 15-day simulation with surface NO₂, NO, CO and O₃ data in the Antwerp region. The modelled daytime NO₂ concentrations are more negatively biased during weekdays than during weekends, indicating a misrepresentation of the weekly temporal profile applied to the emissions, obtained from Crippa et al. (2020). Using a mass-balance approach, we determined a new weekly profile of NO$_x$ emissions, leading to a homogenization of the relative bias among the different weekdays. The ratio of weekend to weekday emissions is significantly lower in this updated profile (0.6) than in the profile based on Crippa et al. (2020) (0.84).

Comparisons with remote sensing observations generally show a good reproduction of the spatial patterns of NO₂ columns by the model. Both APEX and TROPOMI columns are underestimated on the 27/6, whereas no significant bias is found on the 29/6. The two datasets are intercompared by using the model as an intermediate platform to account for differences in vertical sensitivity through the application of averaging kernels. The derived bias of TROPOMI v1.3.1 NO₂ with respect to APEX is



about -10% for columns between $(6\text{-}12) \times 10^{15}$ molec. cm$^{-2}$. The obtained bias for TROPOMI v1.3.1 increases with the $NO_2$ column, following $C_{APEX} = 1.217\, C_{v1.3} - 0.783 \times 10^{15}$ molec. Cm$^{-2}$, in line with previous validation campaigns. The bias is slightly lower for the reprocessed TROPOMI v2.3.1, with $C_{APEX} = 1.055\, C_{PAL} - 0.437 \times 10^{15}$ molec. cm$^{-2}$ (PAL).

Finally, a mass balance approach was used to perform a crude inversion of $NO_x$ emissions, based on 15-day averaged TROPOMI columns. The emission correction is conducted only in regions with high columns and high sensitivity to emission changes, in order to minimize the errors due to wind transport. The results suggest emissions increases over Brussels-Antwerp (+20%), Ruhr Valley (13%), and especially Paris (+39%), and emission decreases above a cluster of power plants in West

Germany.

## 1 Introduction

Nitrogen oxide ($NO_x$ = NO + $NO_2$) pollution is a growing concern in populated, urban areas, due to its adverse effects on human health, ecosystems and the role it plays in further atmospheric processes. In Europe, $NO_x$ pollution sources are largely anthropogenic. Road and non-road transport account for almost half of the total emissions in Europe, while the rest are due to

the energy sector (26%), industrial processes (14%), and small contributions from the residential sector (9%) and agriculture (7%) (Crippa et al., 2018). High $NO_x$ emissions have been linked to premature deaths (Jonson et al., 2017). Environmentally, $NO_x$ pollution can lead to eutrophication of bodies of water, particularly in regions close to emission sources (Stippa et al., 2007). Nitrogen oxides are photochemical precursors of tropospheric ozone (Sillman et al., 1990), which acts as a greenhouse gas with its own environmental and human health impacts (Lelieveld et al., 2015). Ozone production depends equally on the

concentration of volatile organic compound (VOC) species and $NO_x$ (Kleinman, 1994). Thus, the development of accurate and detailed techniques to elucidate the causes of $NO_x$ pollution and predict its consequences is needed to put forward mitigation plans aiming to minimize detrimental effects in the future.

Spaceborne retrievals provide global distributions of key pollutants which cannot be obtained from the sparser, ground-based

air quality networks. Spaceborne measurements of reactive tropospheric pollutants in the UV-Visible range have been in place since the 1990s, with the Global Ozone Monitoring Experiment (GOME) launched in 1995 (Burrows et al., 1999), and its successors the Scanning Imaging Absorption Spectrometer for Atmospheric Chartography (SCIAMACHY) (Bovensmann et al., 1999) and Ozone Monitoring Instrument (OMI) (Levelt et al., 2006; Boersma et al., 2007) launched in the mid-2000s. Each instrument developed on its predecessor, mainly in terms of the spatial resolution (nominally 40 x 320 km$^2$, 30 x 60 km$^2$ and

13 x 24 km$^2$, respectively) at which columns were measured. The TROPOspheric Monitoring Instrument (TROPOMI) aboard the European Space Agency (ESA) Sentinel-5 Precursor (S5P) satellite was developed to capture daily information at even higher resolution, 3.5 x 7 km$^2$ at its launch, improved to 3.5 x 5.5 km$^2$ since August 2019 (Veefkind et al., 2012). Due to the short lifetime of $NO_x$, higher resolution monitoring is critical to capture the spatial and temporal variability of plumes,




especially near urban and industrial regions with strong emission sources. Nevertheless, satellite retrievals have their
limitations and uncertainties, as the observed signal depends on light absorption and scattering over a complex light path
affected by clouds, aerosols, the surface properties and the vertical profile shape of the target gas, all of which are imperfectly
characterized. Thus, satellite measurements must be evaluated against independent data to determine their uncertainties and
biases and to verify their compliance with respect to pre-launch requirements. Numerous validation campaigns were
conducted, generally relying on ground-based or airborne optical measurements (e.g. Judd et al., 2020). Those validation
studies indicated that TROPOMI $NO_2$ columns are negatively biased and that the bias is larger for high columns, although
generally within pre-launch requirements (<50%) (e.g. Griffin et al., 2019; Judd et al., 2020; Zhao et al., 2020; Dimitropoulou
et al., 2020; Chan et al., 2020; Verhoelst et al., 2021; Tack et al., 2021). The most frequent major reasons invoked to explain
the biases are the inadequacy of $NO_2$ profile shapes used in TROPOMI retrievals and the spatial heterogeneity of $NO_2$ fields,
especially near hotspots. The variable extent to which those sources of error are accounted for might explain part of the
differences between biases found in different studies. The profile shape issue is often dealt with by re-calculating TROPOMI
columns using improved profile shapes from a model or from measurements. The issue of spatial heterogeneity can be
addressed through a careful selection of co-location criteria (e.g. Dimitropoulou et al., 2020) or, better, through campaign-
based measurements using airborne remote sensing instruments (van Geffen et al., 2018). In particular, the Airborne Prism
Experiment (APEX) hyperspectral imager was shown to be suitable for TROPOMI validation (Tack et al., 2019), as the satellite
pixels can be fully mapped at high resolution in a relatively short time interval, thereby minimizing the impact of spatial and
temporal mismatches. A dedicated TROPOMI validation campaign was conducted using APEX over Antwerp and Brussels
in June 2019 (Tack et al., 2021).

The region of Antwerp is of special interest, being the most populated municipality of Flanders and an industrial hub housing
the second biggest port in Europe and the second biggest petrochemical cluster in the world. These industries, along with
traffic and shipping emissions, make the Antwerp area a prominent hotspot on spaceborne $NO_2$ maps (Liu et al., 2021).
According to a recent analysis (Flanders Environment Agency, 2017), 13 out of 19 measuring sites in Antwerp showed $NO_x$
concentrations exceeding the European annual limit value. The availability of remote sensing airborne and spaceborne $NO_2$
data as well as of in situ chemical and meteorological observations makes this region especially appropriate for evaluating
regional air quality models. Such models are indispensable tools for testing our knowledge of the processes controlling air
composition and evaluating the impact of mitigation strategies. The performance of those models is however limited due to
various uncertainties in the model parameterizations and, most prominently, in the emissions. With its unprecedented spatial
resolution, TROPOMI offers the promise of providing invaluable information on the distribution of $NO_x$ emissions. The
inverse modelling technique has been used to constrain $NO_x$ emissions based on TROPOMI $NO_2$ data at various scales (e.g.
Lorente et al., 2019; Ding et al., 2020; Souri et al., 2021; Zhu et al., 2021; Botero et al., 2021; Rey-Pommier et al., 2021; Lange
et al., 2022; Fioletov et al., 2022). The characterization of potential biases in TROPOMI $NO_2$ columns is therefore of crucial
importance.



Here we evaluate the Weather Research and Forecasting Model, coupled online with chemistry (WRF-Chem) against a wide array of meteorological and chemical observations in the region of Antwerp and neighboring areas. Those comparisons aim to

assess the model performance and identify the most appropriate setup (choice of model parameterizations and input datasets) for simulating $NO_2$ fields in the area. Next, the model is used as an intercomparison platform for evaluating TROPOMI columns against APEX data.

The WRF-Chem model is described in Sect. 2.1. Section 2.2 presents the combination of global and regional inventories

adopted to specify the emissions, as well as their assumed temporal variations and injection heights. Section 3 describes the observation datasets, including the meteorological and mixing layer height data (Sect. 3.1), the surface in situ chemical measurements (Sect. 3.2), the APEX remote sensing data (Sect. 3.3) and finally the TROPOMI datasets (OFFL v1.3.1 and PAL v2.3.1) (Sect. 3.4). Sections 4.1 and 4.2 present the model comparisons with meteorological and in situ chemical data, respectively. The impact of the boundary layer mixing parameterization on the model performance is also assessed. The

dependence of the model bias on the day of the week is used to propose an improved weekly cycle of anthropogenic emissions in the model. Inter-comparison of WRF-Chem, APEX and TROPOMI $NO_2$ columns is shown in Sect. 4.3. The resulting assessment of TROPOMI biases against APEX data is used to propose a simple bias-correction of TROPOMI columns. A crude inverse modelling method is applied to derive improved emissions over $NO_2$ hotspots in the model domain. Finally, the results are further discussed and put in the perspective of previous validation studies, and conclusions are drawn in Sect. 5.

**2 Model Description and Setup**

**2.1 WRF-Chem**

Weather Research and Forecasting with Chemistry (WRF-Chem) (Grell et al., 2005) is a fully coupled model capable of simulating the chemical processes occurring in the atmosphere simultaneously with meteorology. WRF-Chem model version 4.1.2 and WPS (WRF Preprocessing System) version 4.1 were used, released on July 12th 2019 and April 12th 2019

respectively.

**2.1.1 Model configuration**

The simulation area is centered around Antwerp, the principal region of interest, with two nested domains of 5km x 5km and 1km x 1km resolution, denoted as d01 and d02 respectively in Fig. 1. The projection (Lambert Conformal Conic) and the spatial resolution in the inner domain (1km × 1km) follow the grid definition of the emission inventory for Flanders (see below,

Sect. 2.2.1). The vertical grid has 51 hybrid sigma-pressure levels and extends from the Earth's surface to the model top at 50 hPa. Simulations were conducted for either a short period (~3 days) or a longer period (15 days). The short simulation period covers the two APEX flights and extends from 27/06/2019 00:00UT until 29/06/2019 18:00UT for a total of almost 3 days (66





hours). Each simulation ran on a SGI High Performance Computer using 72 cores, requiring around 30 hours for each short run. Sensitivity simulations suggest only limited deviations of the results when starting the simulation at earlier dates.

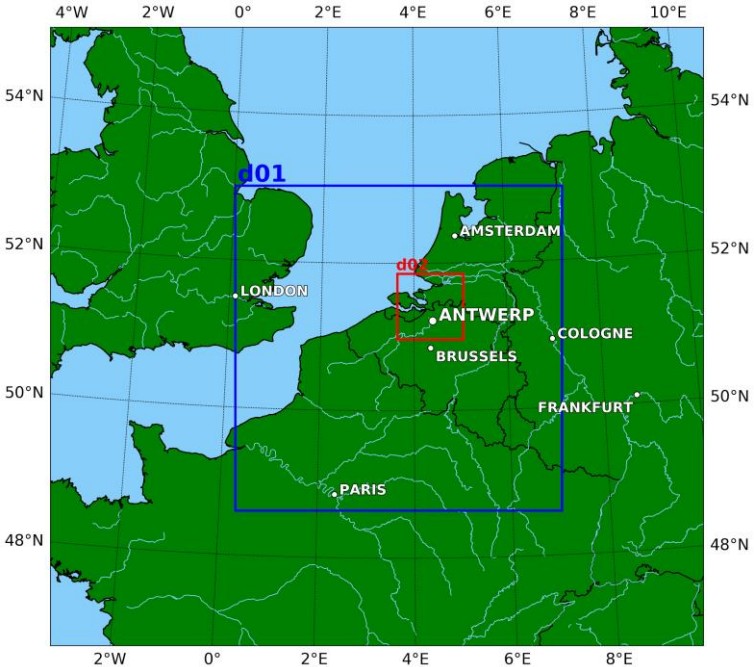


**Figure 1: Map of Western Europe, indicating the two model domains in blue (d01, 5x5 km$^2$ resolution) and red (d02, 1x1km$^2$ resolution).**

Longer (15-day) simulations were also conducted to evaluate the emissions and assess the longer-term model variability

through comparisons with TROPOMI and surface concentration measurements. This simulation period extends from 15/06/2019 00:00UT until 30/06/2019 00:00 UT. Each 15-day run was not set up to run continuously; instead, it consists of a series of partially-overlapping runs of 2 days and 6 hours. Each consecutive run reinitialized the meteorological initial conditions, while the chemical initial conditions were provided by the results of the previous run, except in the case of the initial run at the start of the simulation period.

**2.1.2 Physical parameterizations**

Many options are available in WRF for the parameterizations of physical processes. The basis for the choice of those parameterizations is borrowed from a previous high-resolution WRF-Chem study, conducted in Berlin in 2016 (Kuik et al., 2016), due to the similarity in model set-up and region of interest. The physical parameterization choices utilized in this study are listed in Table 1.

**Table 1. List of WRF-Chem physical parameterizations adopted in this study. The number column refers to the parameterization choice as specified in the WRF-Chem model.**



| Model component | Name | Nr. | Reference |
|---|---|---|---|
| Microphysics | Morrison double-moment scheme | 10 | Morrison et al. (2009) |
| Longwave Radiation | RRTMG scheme | 4 | Iacono et al. (2008) |
| Shortwave Radiation | RRTMG scheme | 4 | Iacono et al. (2008) |
| Land Surface | Noah Land Surface Model | 2 | Tewari et al. (2004) |
| Cumulus Parameterization | Grell-Freitas scheme | 3 | Grell and Freitas (2013) |
| Urban Surface | Single-layer urban canopy model | 1 | Chen et al. (2011) |
| Planetary boundary layer | See Sect. 4 | | |

Due to the importance of planetary boundary layer (PBL) transport processes for the dispersion and vertical distribution of pollutants, the impact of the PBL parameterization on the model results was evaluated through further testing, as detailed in Sect. 4.

### 2.1.3 Chemical mechanism

The Carbon Bond Mechanism Z (CBM-Z) (Zaveri and Peters, 1999) with the Kinetic Pre-Processor (KPP) was chosen to simulate atmospheric gas-phase chemistry, and the Model for Simulating Aerosol Interactions and Chemistry (MOSAIC)
(Zaveri et al., 2008) is adopted for aerosols. The chemical reaction rates of the CBM-Z mechanism were updated in accordance with the latest recommendations of Jet Propulsion Laboratory Publication No. 19-5 (Burkholder et al., 2020). VOC species mapping for the CBM-Z mechanism was done in conformity with previous studies (Chen et al., 2020).

### 2.1.4 Non-emission data

Static geographical data, such as land use category, vegetation and soil type, terrain height, etc., is downloaded from the WRF
users page (https://www2.mmm.ucar.edu/wrf/users/download/get_sources_wps_geog.html/) and is horizontally interpolated using the WRF Preprocessing System (WPS) onto the defined grid. The meteorological boundary and initial conditions for the model are obtained from the NCEP GFS Analysis Products (https://rda.ucar.edu/datasets/ds084.1/). Those are provided as global GRIB2 files at 0.25°×0.25° horizontal resolution and 6h temporal resolution. Chemical boundary and initial conditions are mapped to the WRF grid using species concentrations from the Copernicus Atmosphere Monitoring Service (CAMS)
(Inness et al., 2019) for the species available ($NO_x$, CO, $O_3$, $H_2O_2$, $HNO_3$, $C_2H_6$ and PAN), and CAM-Chem for the remainder




(Emmons et al., 2020). CAMS and CAM-Chem have 0.75°x0.75° and 0.9°x1.25° horizontal resolution, respectively. Both datasets were utilized using a 6-hour temporal resolution.

## 2.2 Emissions

The emissions used in the model simulations originate from multiple datasets, both global and regional. High-resolution
datasets (1x1km$^2$) are adopted for Flanders and the Netherlands, whereas comparatively coarser datasets (0.1°×0.1°) are used over the rest of the domain. Since each dataset has its own specific sector classification, homogenization of the sector types was done to allow for aggregation. The sector types (so-called Selected Nomenclature for Air Pollution, or SNAP, sectors) of the VMM dataset were adopted as the reference, and the sectors in the other inventories were mapped to the SNAP sectors, as illustrated in Table 2 for EDGAR and EMEP. All emissions were processed for model input using the WRF-Chem
preprocessing tool, anthro_emis, provided by NCAR. The monthly-averaged distributions of NO$_x$, CO, NMVOC and SO$_2$ anthropogenic emissions used in both domains of the model are illustrated in Fig. 2.




**Figure 2. Emissions over the two model domains in their respective resolution (average for the period 15/6 - 30/6). The black square is the boundary of the inner domain (d02).**

### 2.2.1 Flanders emission inventory (Flanders Environment Agency, VMM)

The Flemish Institute for Technological Research (VITO) processed emissions for CO, $NH_3$, total NMVOC, $NO_x$ (as $NO_2$), $PM_{10}$ and $SO_x$ (as $SO_2$) over Flanders for 2017, split over 10 sectors, originating from the Flanders Environment Agency (VMM), (Flanders Environmental Agency (VMM), 2017). This inventory contains both gridded emissions over 1x1km$^2$ and point source emissions, corresponding to the industrial sector.



### 2.2.2 Dutch emission inventory

High resolution emissions over the Netherlands were obtained from the Government of the Netherlands Pollutant Release and Transfer Register for all species in the VMM inventory (http://www.emissieregistratie.nl, last accessed June 2020). These are $1x1km^2$ resolution estimates representing the yearly total for 2017, split by sector and sub-sector. The species include $NO_2$, CO, $NH_3$, NMVOCs, $PM_{10}$ and $SO_2$. The data was regridded from its original projection into the VMM projection. The specifications of the VMM and Dutch emission projections were obtained from their corresponding shape files in QGIS. The horizontal coordinates of the Dutch inventory cells were reprojected into the VMM grid using the Proj and transform functions from the pyproj Python module (https://github.com/pyproj4/pyproj).

### 2.2.3 EDGAR V4.3.2

The Emissions Database for Global Atmospheric Research (EDGAR) provides global sector-specific anthropogenic emissions on a 0.1° x 0.1° spatial grid. In this study, two different dataset versions have been used. From EDGAR v4.3.2 (Crippa et al., 2018; Huang et al., 2017), we use the annual disaggregated emissions of 25 NMVOC species and classes. The most recent year in this dataset is 2012, and this was used for the NMVOCs. Since only the total NMVOC emissions were available from the Flemish and Dutch inventories, the NMVOC speciation among different NMVOCs was obtained from EDGARv4.3.2 and combined with the high-resolution total NMVOC data over Flanders and the Netherlands.

### 2.2.4 EDGAR V5

Black carbon (BC) and organic carbon (OC) emissions were taken from EDGAR V5.0 2015 (Crippa et al., 2020), which are monthly-specific. These emissions were utilized over both domains, and regridded to the desired spatial resolution using the WRF-Chem preprocessor anthro_emis.

### 2.2.5 EMEP

Emissions from the European Monitoring and Evaluation Programme (EMEP, https://www.ceip.at/the-emep-grid/gridded-emissions) were used for $NO_x$, CO, $PM_{2.5}$, $PM_{10}$ and $SO_x$ over Europe. These are yearly emissions gridded at 0.1° x 0.1° from 2012, provided as both a total emission and split by individual sector. Total NMVOC emissions are also available from EMEP, but were not used as they are not speciated.

**Table 2. Correspondence between the emission sectors of the SNAP, EDGAR and EMEP inventories, as adopted in this work.**

| VITO CATEGORIES (SNAP) | EDGAR | EMEP |
|---|---|---|
| SNAP1: Combustion and Energy in Transformation Industry | Energy Industry (ENE) | Public power |



| SNAP2: Non-industrial combustion plants | Residential (RCO) | Other stationary combustion |
|---|---|---|
| SNAP3: Combustion in manufacturing industry | Combustion in manufacturing industry (IND) | Industry |
| SNAP4: Industrial processes | | |
| SNAP5: Extraction and Distribution of Fossil Fuels and Geothermal Energy | Fuel production/transmission (PRO) | |
| SNAP6: Solvent and Other Product Use | Application of solvents (SOL) | Solvents, fugitive |
| SNAP7: Road Transport | Road Transport (TRO) | Road transport |
| SNAP8: Other Mobile Sources and Machinery | Non-road transport (TNR) | Aviation, offroad, shipping |
| SNAP9: Waste Treatment and Disposal | Waste water (WWT) | waste |
| SNAP10: Agriculture | Agricultural soils (AGS) | Agrilivestock, agriother |


The total annual emissions in the model domain amount to 770 Gg(NO), 1960 Gg (CO) and 2170 Gg (NMVOC). Figure 3 shows the sector contributions for the total $NO_x$, CO and NMVOC emissions over the entire model domain. The "other" category is a sum of the least contributing sectors (less than 1% of the total emission). For $NO_x$, the dominant sectors are the SNAP sectors 7 (road transport) and 8 (mostly shipping), followed by the energy sector (SNAP1). For CO, industry (SNAP4) accounts for almost half of the total, while the transport (7+8) and residential (2) sectors account for most of the remainder. Contributions to NMVOC emissions come from all SNAP sectors, among which SNAP6 (solvents) is largely dominant, as well as from biogenic sources, more specifically biogenic isoprene that accounts for 23% of total NMVOC emissions.



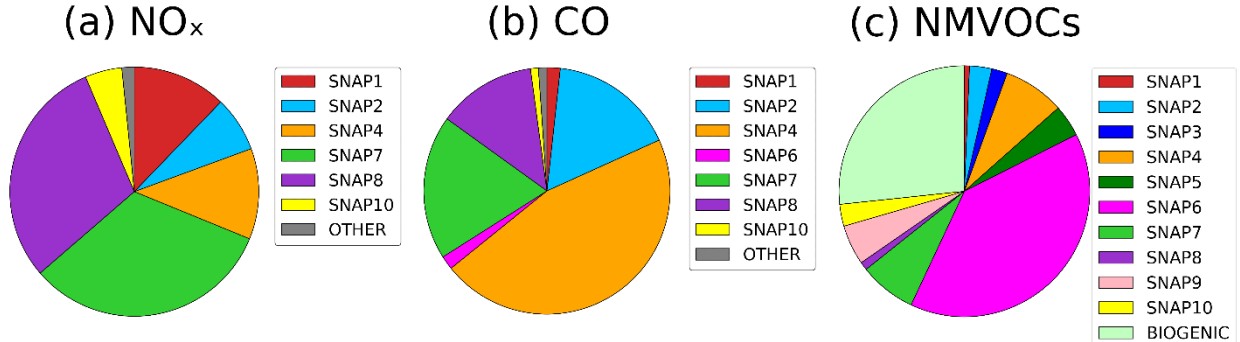

**Figure 3. Sector contribution (%) to total emission for NO$_x$ (as NO), NMVOCs and CO. Sectors with less than 1% contribution are**
**grouped into the "OTHER" category, which differs between species.**

### 2.2.6 Injection heights

Industrial emissions from the VMM inventory have vertical distribution information, namely the injection height, which ranges between 0 and 204m. For other emission categories (except aircraft) and outside of Flanders, emissions are assumed to occur at surface level. The injection height information is used to populate the 3 first vertical levels of the emission input files
(approximately 0-50m, 50-110m, 100-200m above ground level). About 94% of NO$_x$ emissions from Flanders is injected above the surface in the VMM inventory, but only 11% is injected above the first model layer (0-50m) (Sessions et al., 2011).

### 2.2.7 Aircraft and lightning

Global NO$_x$ emissions from aircraft are provided from the CAMS-GLOB-AIR inventory (Granier et al., 2019), which is based on CEDS aircraft emission data (Hoesly et al., 2018). The data has 0.5° x 0.5° horizontal resolution and monthly variation,
with 25 vertical levels between the surface and 15km altitude. The dataset used in this study is for the year 2019. To avoid double-counting, CAMS-GLOB-AIR emissions at the first level (closest to the surface) were omitted, since surface-level aircraft emissions are accounted for in the surface emission inventories.

Lightning-generated NO$_x$ (LNOx) is computed within the WRF-Chem model through the additional physics parameterization
for the lightning process, based on the PR92 scheme (Price and Rind, 1992). The amount of LNOx is determined from the lightning flash rate (parameterized based on the convective cloud top height calculated by WRF), with different formulations for continental and marine thunderstorms.

### 2.2.8 Temporal variation of anthropogenic emissions

The surface emission inventories described in Sections 2.2.1–2.2.5 generally ignore seasonal, weekly and diurnal variations of
anthropogenic emissions, which can however be significant. Diurnal, weekly and seasonal cycles of emissions were included in the model based on the detailed sector-specific and country-specific temporal profiles of Crippa et al. (2020).



For each sector, the emission at any time is obtained by multiplying the annual emission by factors representing their dependence on the hour of the day, day of the week and month of the year, following Crippa et al. (2020), as described by Eq. 1. The temporal profiles pertain to the EDGAR sectors. The correspondence between the EDGAR and SNAP sector categories is displayed in Table 2.

$$TF = \alpha_{s,d,h} \times \beta_{s,w} \times \gamma_s \tag{1}$$

Where TF represents the temporal factor made up of its three components: a diurnal factor, $\alpha$, dependent on sector, day type (weekday, Saturday or Sunday) and hour, represented by subscripts s, d and h, respectively. $\beta$ stands for the daily factor, dependent on the sector (s) and day of the week (w). The final component of the temporal factor is $\gamma$, corresponding to a monthly factor only dependent on the sector. Each component in the temporal factor is country-dependent. We adopted the temporal profiles provided for Belgium, which are very similar to those for neighboring countries. Each of the temporal variations is also monthly-dependent. Simulations were conducted in the month of June. Temporal variation was not applied to the aircraft emissions, as these do not correspond to an EDGAR category.

Figure 4 shows the temporal features of different sectors over the last two days of the 3-day runs, namely the 28 and 29 of June 2019 corresponding to a Friday and a Saturday. The time-series shows a distinct diurnal cycle for each of the categories, as well as distinctly lower emissions on the 29th due to the week-end effect. SNAP9 is not shown in this figure, as it has no temporal variation.

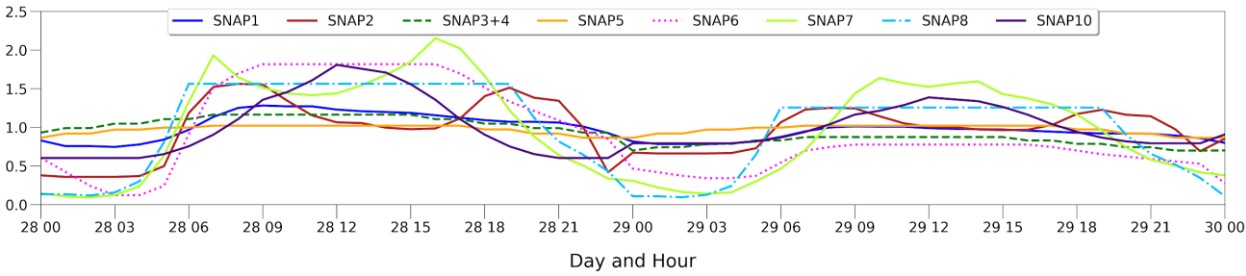

**Figure 4. Temporal profile of SNAP categories over Friday (28/06) and Saturday (29/06), showing a variation in diurnal shape for the different sectors and over the two days, time indicated as local time or UTC+2.**

The temporal profile of road transport (SNAP7) shows a minimum at early hours of the day on both Friday and Saturday, as well as two distinct peaks on Friday, corresponding to the rush hours between 6 and 8 AM and 3 and 5 PM. Non-road transport (SNAP8) shows a constant value during the daytime for both Friday and Saturday. The shape for the non-industrial combustion sector (SNAP2) shows two broad peaks on both days between 6 and 11 AM and 5 and 11 PM, corresponding to hours before and after work, when there is more activity within the home. The industrial sectors (SNAP1, SNAP3+4, SNAP5 and SNAP6) generally have flatter curves, indicating a more constant release of emissions throughout the day.





### 2.2.9 Biogenic emissions

270

Biogenic emissions of isoprene are calculated on-line in WRF-Chem using the algorithms of the Model of Emissions of Gases and Aerosols from Nature (MEGAN) (Guenther et al., 2012). The emissions depend on meteorology (as calculated by WRF) and on emission factors and land use/land cover parameters provided as input at approximately 1km resolution. Biogenic isoprene emissions total 676.5 Gg yr$^{-1}$ over the two domains.

275

### 3 Methodology

#### 3.1 Meteorological data

##### 3.1.1 Ground-based

The Royal Meteorological Institute of Belgium (RMI) operates a network of automatic weather stations over the Belgian territory, recording near-surface meteorological observations, specifically temperature, relative humidity, air pressure,

280 precipitation, global solar irradiance, wind speed and wind direction. The measurements are obtained automatically every hour. Data was acquired for the month of June 2019 for two stations in the Antwerp area, Stabroek (51.3493° N, 4.3789° E) and Sint-Katelijne-Waver (51.0696° N, 4.5346° E). The location of the stations is indicated in Fig. 5. Both stations are within the inner domain, within the Antwerp area (https://www.meteo.be/en/about-rmi/observation-network/automatische-weerstations).



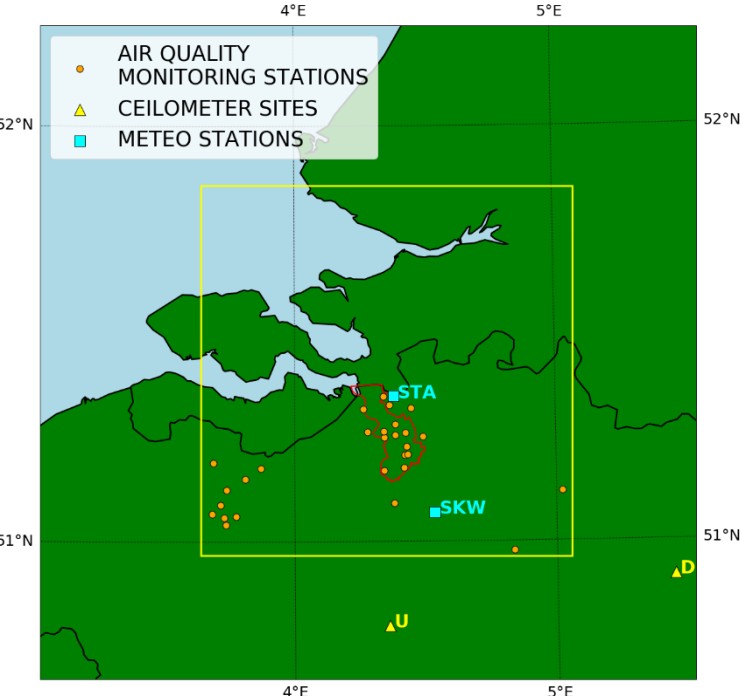

**Figure 5. Map showing the geographical location of different in-situ data measuring sites. The meteorological (blue square) and air quality (orange circle) sites are in the innermost domain, indicated by the yellow square, while the ceilometer stations (yellow triangle) are in the surrounding area. The red outline represents the Antwerp municipality.**

### 3.1.2 Radio- and ozonesonde

Ozonesondes are balloon-borne instruments measuring ozone concentration as they travel from the surface to altitudes of around 30km, in the mid-stratosphere. The type of ozonesonde currently used by the RMI at the Uccle station (50°48'N, 4°21'E, 100 m above sea level, indicated by "U" in Fig. 5) is an Electrochemical Concentration Cell (ECC) sonde, which measures ozone concentration through a reaction with ambient air in an electrochemical cell that generates an electric current proportional to the amount of ozone in the air (Van Malderen et al., 2021; Deshler et al., 2017). These are systematic measurements conducted roughly every 3 days. We compared the model with sonde data for the 28[th] of June. Alongside ozone, air temperature, relative humidity, wind direction and wind speed information are measured with the radiosonde the ozonesonde is coupled with. Those airborne instruments take measurements during its ascent to and descent from the maximum altitude, and the horizontal coordinate path of the balloon can be reconstructed using the measured wind speed, wind direction and time passed from the start of measurements.

### 3.1.3 Ceilometer

There are four Automatic LIDAR (Light Detection and Ranging) Ceilometer (ALC) monitoring stations in Belgium, set up by the RMI to measure cloud base height and mixing layer height (https://ozone.meteo.be/instruments-and-observation-





techniques/lidar-ceilometer). The Vaisala CL51 instrument emits a single-wavelength pulse vertically into the atmosphere, and measures the time taken for a backscattered signal to return, proportional to the size and scattering cross section of molecules and particles (Haeffelin et al., 2016). The PBLH is retrieved from ceilometer measurements by applying a similar

approach used in Pal et al. (2013). The retrieval algorithm is based on the 2D gradients method (Haeffelin et al. 2012) in combination with the variance method (Menut et al. 1999). The retrieved PBLH are available for the month of June 2019 at 10 minutes intervals and are checked for quality following the methodology used in de Haij et al. (2007). In case of rain/fog, the retrieved PBLH is not used. For all other sky conditions, a quality criteria equal to 0.85 was used, a value higher than 0.85 is an indication of an incorrect retrieved PBLH. We use measurements obtained at the stations of Uccle (50.7975°N, 4.3594°E)

and Diepenbeek (50.9155°, 5.4503°). These stations are indicated in Fig. 5, where each station is labeled by its initial.

**3.2 In situ surface chemical observations**

The Belgian Interregional Environmental Agency (IRCEL-CELINE) hosts air quality measurement stations, providing timeseries of hourly measurements of common pollutant concentrations ($NO_2$, $NO$, $CO$ and $O_3$). The 27 stations used for evaluation of the model are those located within the inner model domain (Figure 5), listed in Table S1. When comparing the

modelled $NO_2$ concentration with station data, a correction factor ($F_{int}$) is applied (either to the observations or to the model results) to correct for the known existence of interferences in the chemiluminescence $NO_2$ measurement due to $NO_y$ reservoir compounds including $HNO_3$ and PAN (Lamsal et al., 2008). More precisely, in most instances, the modelled $NO_2$ concentrations are multiplied by the correction factor ($F_{int}$) given by

$$F_{int} = 1 + \frac{0.95 \times [PAN] + 0.35 \times [HNO_3]}{[NO_2]} \qquad (2)$$

where $[NO_2]$, $[PAN]$ and $[HNO_3]$ are the modelled mixing ratios of $NO_2$, PAN and $HNO_3$. In other words, the measurements are being compared to interference-corrected $NO_2$ concentrations, hereafter denoted $NO_2^*$. Alternatively, the modelled $NO_2$ could be compared to the measured concentrations divided by $F_{int}$, in order to remove the estimated interference contribution from the measurements. Because the correction $F_{int}$ involves model-calculated concentrations, however, this procedure is inappropriate when evaluating multiple model runs against $NO_2$ data.

**3.3 APEX**

$NO_2$ tropospheric vertical column densities (VCDs) were retrieved over Antwerp during two flights utilizing hyperspectral Airborne Prism EXperiment (APEX) observations, as part of the S5P validation campaign over Belgium (S5PVAL-BE). The APEX instrument is a pushbroom hyperspectral imager that integrates spectroscopy and 2-D spatial mapping in high-resolution (~ 75m x 120m). APEX utilizes backscattered solar radiation over a wavelength range of 370 to 2540 nanometres. The flights

over Antwerp took place on the 27[th] and 29[th] of June 2019, using the APEX instrument aboard a Cessna 208B Grand Caravan EX at an altitude of 6.5km above ground level. The days on which the flights took place were chosen for their good visibility




due to cloud-free conditions. The flight times were chosen to coincide within 1 hour of the times of the S5P overpasses on the corresponding days. Each APEX $NO_2$ column is provided with vertically-resolved box air mass factors (AMFs) estimated from radiative transfer calculations. The total AMF, equal to the ratio of the slant column to the vertical column, is computed

by vertically integrating the box AMFs along the a priori $NO_2$ profile. This profile is taken to be a constant mixing ratio in the PBL and zero in the free troposphere, the PBL height being obtained from ceilometer data in Uccle, near Brussels (Tack et al., 2017; 2021). The averaging kernels ($A_l$, with $l$ the vertical level) required for comparison of model data with APEX columns are calculated as the ratio of the box AMFs to the total AMF (Eskes and Boersma, 2003). APEX columns and averaging kernels are regridded at the 1 $km^2$ WRF resolution. The model $NO_2$ columns are computed by vertically integrating the model partial

columns (below the tropopause and mapped onto the APEX vertical grid) multiplied by the averaging kernels.

## 3.4 TROPOMI

The TROPOspheric Monitoring Instrument (TROPOMI) was launched aboard the European Space Agency (ESA) S5P satellite in 2017 to monitor and quantify air quality across the globe (Veefkind et al., 2012). S5P is a near-polar, sun-synchronous satellite, with a 13:30h local overpass time. TROPOMI is a nadir-viewing push-broom imaging spectrometer with ultraviolet

(UV), visible (VIS), near-infrared (NIR) and shortwave infrared (SWIR) spectral bands, which allow for measuring atmospheric constituents such as nitrogen dioxide ($NO_2$), ozone ($O_3$), carbon monoxide (CO) and other compounds at the high spatial resolution of 7km x 3.5km (5.5km x 3.5km since August 2019). The retrieval of tropospheric $NO_2$ is a three step process. Firstly, a Differential Optical Absorption Spectroscopy (DOAS) method is used to obtain the total slant column density of $NO_2$ from the Level-1b radiance and irradiance spectra measured by TROPOMI. This method utilizes a spectral range of 405-465

nm and is based on the non-linear fitting approach for OMI (van Geffen et al., 2020). The second step requires a separation of tropospheric and stratospheric $NO_2$, realized using data assimilation of slant columns with the TM5-MP chemistry-transport model (Williams et al., 2017). Finally, the tropospheric slant column density, derived in the previous step, is converted to a tropospheric vertical column density using pre-calculated air-mass factors (AMFs). AMFs are obtained from radiative transfer calculations using $NO_2$ vertical profiles from the TM5-MP chemistry-transport model. The $NO_2$ retrieval and the individual

steps are described in further detail in the TROPOMI ATBD (Algorithm Theoretical Basis Document) of the total and tropospheric $NO_2$ data products (van Geffen et al., 2018). To enable the comparison between TROPOMI and WRF-Chem, the model $NO_2$ columns are computed at the WRF-Chem resolution by convolution of the modelled tropospheric partial columns with the TROPOMI averaging kernels. The resulting columns are then regridded onto the TROPOMI resolution, using the latitude and longitude coordinates of the four corners of each TROPOMI cell. The quality filter recommended by the

TROPOMI ATBD is applied to both model and measurements, i.e. only pixels with QF > 0.75 are kept for further analysis. We present model evaluations against the standard L2 tropospheric $NO_2$ product (OFFL v1.3.1) as well as against the newly released PAL reprocessing based on version 2.3.1 of the operational processor (https://data-portal.s5p-pal.com/product-docs/no2/PAL_reprocessing_NO2_v02.03.01_20211215.pdf; van Geffen et al., 2022) available for download on the S5P-PAL





noted TROPOMI_v1.3 and TROPOMI_PAL, respectively.

## 4 Results

Given the important role of the PBL parameterization in the simulation of atmospheric composition, the PBL scheme was varied in the evaluation of WRF-Chem performance against measurements. Ten PBL schemes were tested, listed in Table 3. The shorthand notation refers to the model simulation utilizing the corresponding PBL scheme. The P4 run crashed before

completion for reasons unknown (stopped around 04:30 on the 29th), which is why the data series is incomplete on multiple plots. The choices of the PBL scheme and surface layer parameterization are coupled, and the suitable pairs are suggested in the WRF User Guide V4 (https://www2.mmm.ucar.edu/wrf/users/docs/user_guide_v4/contents.html). When several choices are possible for a given PBL scheme, the effect of the choice of surface layer scheme was tested through comparison with meteorological data. The results indicate very little sensitivity to the choice of surface layer scheme. Surface layer option 1

(Revised MM5 Monin-Obukhov scheme, Jiménez et al., 2012) was selected in cases where multiple choices were proposed, in order to ensure the best consistency amongst the different runs. The surface layer scheme options chosen for each simulation are listed in Table 3.

In all comparisons shown in this section, the time refers to local time, i.e. UTC + 2.

**Table 3. List of WRF-Chem planetary boundary layer parameterizations and their coupled surface layer scheme tested in this study. The number in the last column identifies the surface layer scheme, as defined in the WRF documentation (Skamarock et al. 2019). Note that schemes 7 and 99 were not tested as scheme 7 appears incompatible with the other physical parameterizations used and scheme 99 will be removed in future versions of the model.**

| Short-hand | Scheme name | Reference for PBL scheme | Coupled surface layer scheme |
|---|---|---|---|
| P1 | Yonsei University Scheme (YSU) | Hong et al. (2006) | 1 |
| P2 | Mellor–Yamada–Janjic Scheme (MYJ) | Mesinger, 1993 Janjić (1994) | 2 |
| P4 | Quasi–normal Scale Elimination (QNSE) Scheme | Sukoriansky et al. (2005) | 4 |
| P5 | Mellor–Yamada Nakanishi Niino (MYNN) Level 2.5 Scheme | Nakanishi and Niino (2006) | 1 |





| P6 | Mellor–Yamada Nakanishi Niino (MYNN) Level 3 Scheme | Nakanishi and Niino (2006) | 5 |
|----|----|----|----|
| P8 | Bougeault–Lacarrere Scheme (BouLac) | Bougeault and Lacarrere (1989) | 1 |
| P9 | University of Washington (TKE) Boundary Layer Scheme | Bretherton and Park (2009) | 1 |
| P10 | TEMF Scheme | Angevine et al. (2010) | 10 |
| P11 | Shin-Hong Scale–aware Scheme | Shin and Hong (2015) | 1 |
| P12 | Grenier–Bretherton–McCaa Scheme | Grenier and Bretherton (2001) | 1 |

## 4.1 Comparisons with meteorological observations

### 385 4.1.1 Ceilometer

The PBL height (PBLH) from WRF-Chem was compared with ceilometer measurements from Uccle and Diepenbeek (labeled U and D on Fig. 5) over the 3-day simulation period. The retrieved PBLH are split into two categories based on the quality criteria. They are shown on Fig. 6 as green dots (good quality data, QF < 0.85) and brown dots (low quality).The measured data and corresponding model output were averaged over the two locations.

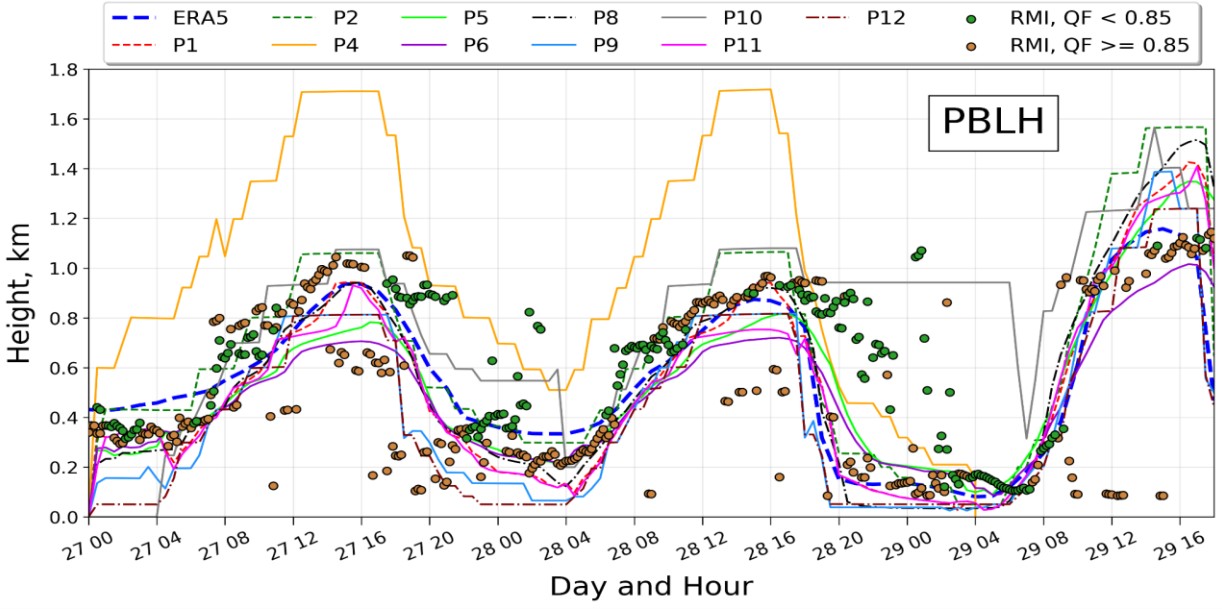






**Figure 6. Time series of measured and modelled PBL height between 27/6 and 29/6/2019 (average of 2 sites). Good-quality measurements (quality filter QF < 0.85) are shown as green dots, low-quality data (QF > 0.85) as brown dots. The dark blue dashed line is the PBL height from the ERA5 reanalysis (C3S, 2017). The other curves represent the WRF model output for each of the boundary layer schemes listed in Table 3.**

Generally, the different PBL parameterization runs are able to reproduce the observed temporal shape of PBLH, especially during the daytime hours (8AM-6PM). All schemes perform well regarding the time (~16PM) and approximate magnitude (1km) of the maximum PBLH. However, most schemes underestimate the PBL height during the nighttime (8PM-4AM), especially when comparing with the good quality ceilometer data. The extent of the underestimation varies between the different PBL schemes. P10 is a noticeable exception, as it consistently overestimates the observations during the night. Both

P9 and P12 predict a sharp drop in PBLH around 18PM (not seen in the observations) and exhibit the lowest PBL height for both nights. However, the relatively high observed PBLH in the late afternoon might be an artefact due to the persistence of a residual aerosol layer which does not subside rapidly despite the weakening of turbulence at that time (Haeffelin et al., 2012). There are clear outliers among the different schemes – P4 performs the worst as its PBLH is exceedingly higher than the measurements. Besides P9 and P12 (of which poor performance might partly due to the limited reliability of ceilometer data

in late afternoon), P4 and P10 exhibit the poorest statistics (correlation, RMSE, mean bias) among the different schemes (see Table S2 in the Supplement).

### 4.1.2 Surface meteorology

Figure 7 presents the time series of meteorological parameters observed and simulated by the model in the region of Antwerp (average of two stations, Stabroek and Sint-Katelijne-Waver). The model follows the observed diurnal shape of surface

temperature very well, with Pearson's $R^2$ values above 0.98 for every simulation except P6 (too cold in the afternoon and night) and P10 (too warm in the morning). Nighttime temperature from WRF-Chem is slightly but consistently underestimated by all schemes, whereas the agreement between the modelled and observed temperature is generally excellent during the daytime. P6 and P10 are among the worst performing PBL schemes for correlation, RMSE and mean bias (Table S2). P4 also displays a large negative bias.

Similarly to temperature, the measured relative humidity exhibits a diurnal shape that WRF-Chem is able to simulate, with a maximum during the nighttime, and lower values during the day. As for temperature, P4, P6 and P10 are the worst-performing runs. Those three schemes overestimate the relative humidity, especially P6 during the day (by more than 10%) and P10 during the night. P9 is consistently too dry.

The observed solar irradiance reveals essentially clear-sky conditions during the three days. This is well-reproduced by most

PBL runs, except P6, which underestimates the solar irradiance between 9 and 12PM on the first two days, likely due to cloudiness and in line with the overly moist conditions calculated by this scheme. To a lesser extent, P5 also slightly underestimates the solar irradiance on the 28th between 6 and 12PM.





Among the meteorological variables, wind speed displays the most variability between the different PBL runs, as well as the
highest discrepancy with the measurements.  Wind speed is overestimated by all schemes. The overestimation is highest for
P2 and P4, while P10 also performs poorly for both wind speed and wind direction. The other schemes perform similarly, with
a moderate wind speed bias of ~1.1 m/s and a correlation of ~0.85 for wind direction. Similar wind speed overestimations have
been noted in previous WRF-Chem evaluation studies over urban areas (e.g. Feng et al., 2016; Kim et al., 2013). Note that the
single-layer Urban Canopy Model (UCM) used in our study as urban surface model (Table 1) has been found by those studies
to provide the best results, based on comparisons with ceilometer and wind measurements over cities. Overall, those
comparisons for the surface meteorological variables indicate a good model performance, except for the few outliers that
consistently perform poorly, mainly P4, P6 and P10.





**Figure 7. Observed and modelled evolution of surface meteorological parameters on 27-29/6/2019. (a) Temperature, (b) relative humidity, (c) solar irradiance, (d) wind speed and (e) wind direction. The dark blue dashed line represents the observations, averaged**




over the two measuring sites, while the other lines represent the output from the model sensitivity runs, labeled in the legend (see Table 3).

### 4.1.3 Sonde

Variation in the output of the PBL runs occurs only in the lowest part of the troposphere (<1200m). The meteorological output of the different schemes is very similar at higher altitudes. The model succeeds very well in reproducing the vertical profile of meteorological parameters in the free troposphere (Fig. 8), especially temperature and the winds. The model underestimates ozone between 3 and 10 km altitude, but agrees very well with the sonde in the lowest layers. To a large extent, the discrepancy in the upper troposphere is due to underestimated ozone mixing ratios by the CAMS analysis for this date (also shown on Fig.

8(e)). CAMS ozone is used to specify the initial and lateral boundary conditions in the model.

The impact of the PBL option on the comparisons between the simulated and measured data is examined below approximately 1200m altitude. Both sonde ascent and descent are considered — the average of the two is plotted in Fig. 9 and the combined data is used to calculate the statistics in Table S2. Most of the runs slightly overestimate the temperature in the lowermost

600m. As with the surface data, P6 is an outlier in that it displays the highest, negative mean bias. P8 and P10 have the highest positive biases. These findings are consistent with the surface temperature comparison.

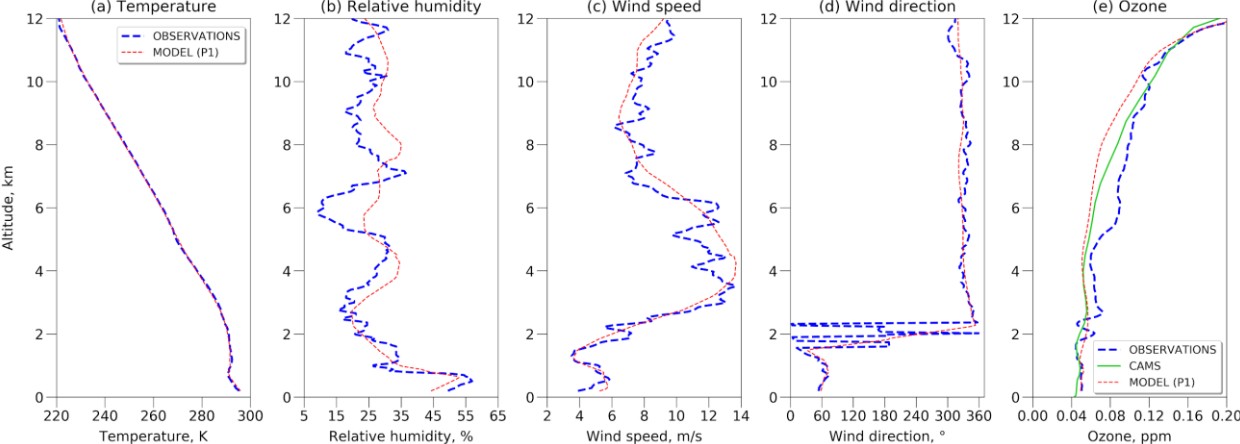

**Figure 8. Vertical profile of meteorological parameters and ozone mixing ratio measured by ozonesonde on 28/6/2019 (dashed blue line) and calculated by WRF-Chem (red dotted line), shown between 0 and 12km (approximate tropopause height) for one PBL**
**scheme (P1). The ozone profile from the CAMS analysis (green) is also shown in panel (e).**





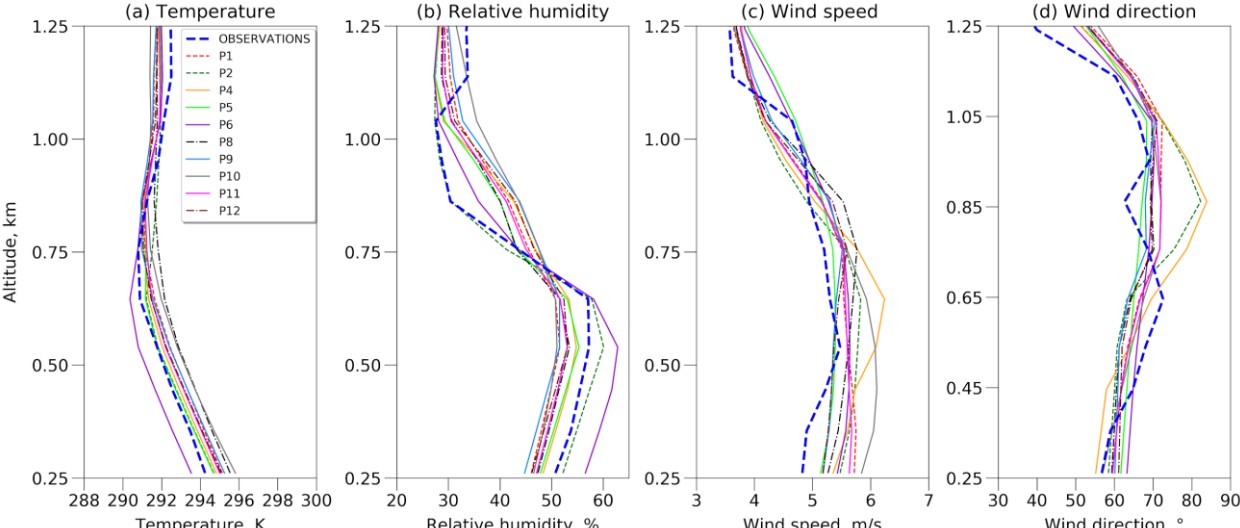

**Figure 9. Vertical profile of meteorological parameters (thick blue line) observed by radiosonde on 28/6/2019 below 1.25 km altitude and WRF-Chem results for different boundary layer parameterizations (see Table 3).**

Although all runs follow the general shape of the relative humidity measured profile, the vertical gradient between 700 and 1000m is generally underestimated by the model. P2 is the exception, which provides the best overall agreement with the measurements (Table S2). The largest biases are found for P6 (too moist in the lowest layers) and for P10 (highest overestimation above 800m).

All PBL schemes overestimate the wind speed at near-surface altitudes, in consistency with the surface data. P2 and P4 show a large positive bias in wind direction (ca. +20°) around 900m altitude and are generally the worst-performing schemes for both wind speed and wind direction, with the lowest correlations and highest RMSE values. P8 shows the best performance statistically, despite being amongst the worst performing options for the surface wind speed. Besides P2 and P4, most of the schemes display similar statistics (Table S2).

### 4.2 Comparisons with surface chemical observations

#### 4.2.1 Role of PBL scheme on model comparison with surface NO₂ data

The output of the test runs was further compared against measured surface $NO_2$ at the IRCEL-CELINE stations located in the inner domain. The modelled values shown in Fig. 10 have been corrected to account for the interference of other $NO_y$ species in the measurements, as described in Sect. 3.2. To help interpreting these comparisons, Fig. 11 displays the vertical profiles of modelled $NO_2$ mixing ratios in the early afternoon (13h30 on the 27[th]) and during the night (1h30 on the 28[th]), below 1.4km altitude. Note that the choice of PBL scheme has essentially no impact above that altitude.




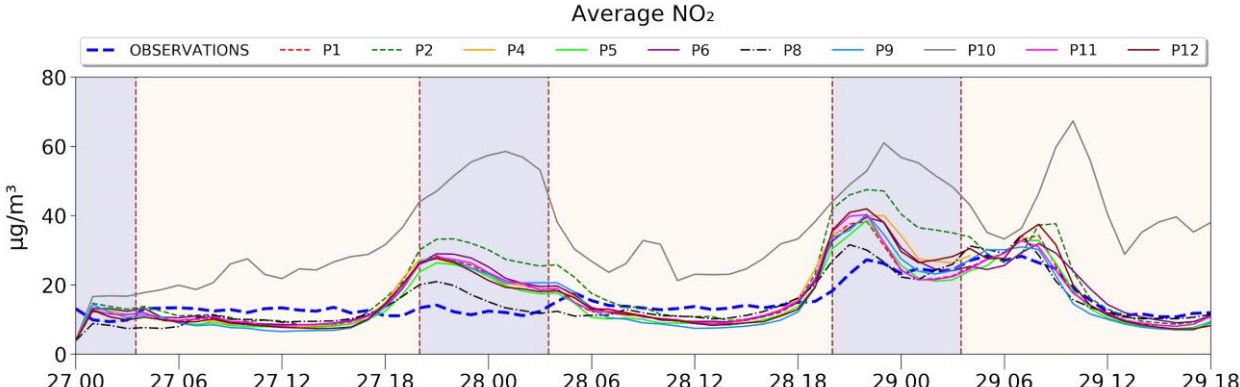

**Figure 10. Evaluation of modelled NO₂\* concentrations (corrected to account for interference of other NOᵧ species, see text) using station data over the 3-day simulation period. Both the model data and observations are averages over 24 stations within the inner domain. The thick dashed line represents the measurements while the other curves represent the different PBL runs. The vertical dashed lines separate night and daytime, shaded in blue and orange, respectively.**


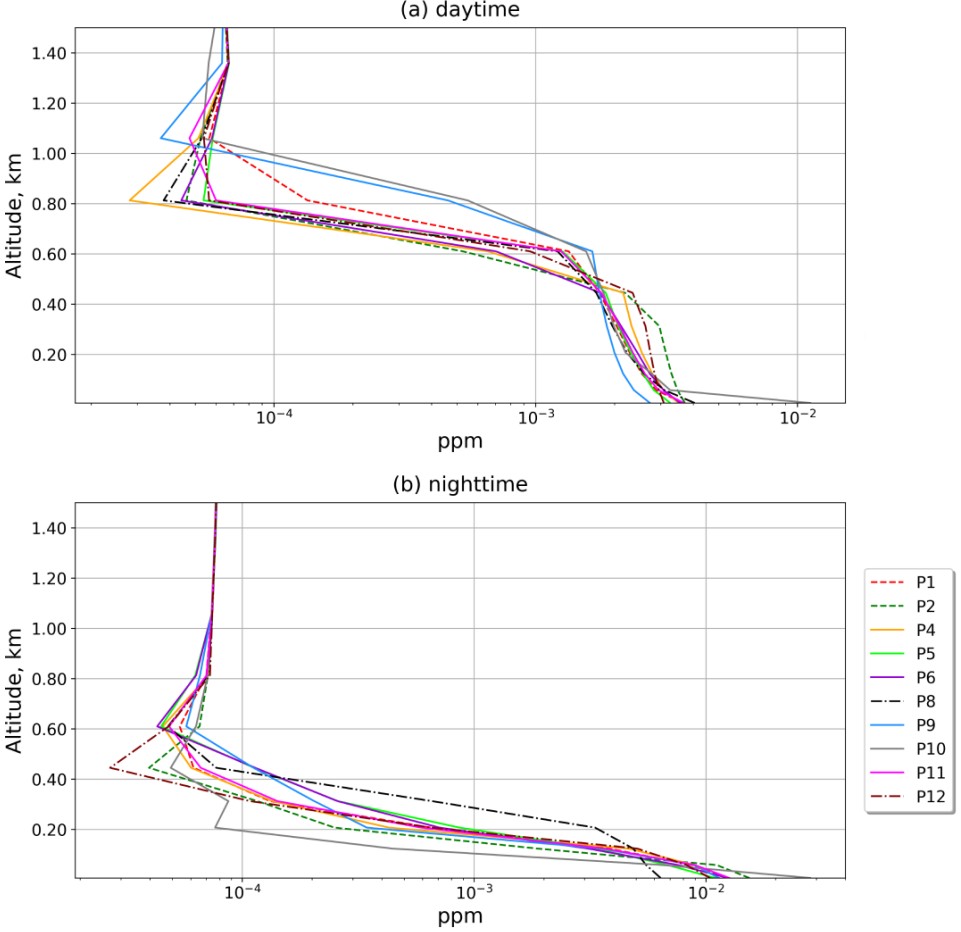




**Figure 11. Average NO₂ vertical profile over the APEX region, shown for all sensitivity simulations, below 1.4km altitude (a) at 1:30PM on 27/6, and (b) at 1:30AM on 28/6. Above 1.4km, differences between model sensitivity runs are negligible.**

Similarly to the comparison with meteorological parameters, most of the PBL options are cohesive in their performance against observations, except for four outliers. P10 consistently overestimates NO₂ over the three-day period by a factor of ~2. This overestimation is associated with a very steep vertical gradient of NO₂ concentration within the PBL (see Fig. 11) which suggests insufficient boundary layer turbulent mixing. The schemes P2 and especially P4 lead to high RMSE and MB values (Table S3), consistent with their relatively lower performance at simulating meteorology, in particular the wind. P8 exhibits

the least-pronounced diurnal cycle of NO₂ concentration, in better agreement with the observations than the other simulations. Excluding P4 and P10, surface NO₂ exhibits an average bias between -4.3 and -25% during daytime (6AM-8PM) and overestimations of 8.2-77% for nighttime concentrations. On the 27th and 28th, all remaining schemes generally underestimate NO₂ concentrations during daytime hours and overestimate NO₂ at night, highlighting a potential issue with the diurnal profile of NO₂ in the model. Possible causes for this pattern, including issues with PBL vertical transport and with the chemical sinks

of NO₂, will be discussed in the next subsections. On the 29th, however, most schemes perform very well. The larger daytime biases on weekdays (27-28th) compared to Saturday (29th) for all runs (except P10) suggest an issue with the weekly cycle of emissions, which is further explored in Sect. 4.2.5. Based on these results and on the comparisons with meteorological measurements, the PBL options P2 (MYJ), P4 (QNSE), P6 (MYNN Level 3) and P10 (TEMF) are considered less reliable for simulating air composition over Antwerp and surrounding areas, and are therefore not recommended.


Due to computational resource limitations, we adopt only one PBL option (P1, the YSU scheme) for the 15-day simulations. The choice is justified by the good performance of the P1 simulation across all meteorological comparisons. In addition, it compares similarly to most other schemes against surface NO₂ data, when excluding the few outliers noted above. The P8 scheme (BouLac) could have been a meaningful alternative choice given its better agreement against surface NO₂ data. Note

however that the lower amplitude of the diurnal cycle of surface NO₂ simulated with P8 might be partly due to the near-absence of diurnal cycle in wind speed obtained with that scheme (Fig. 7), whereas the observations and most other model simulations indicate higher wind speeds during the day than during the night. Since high wind speeds cause faster export and dilution of pollution plumes, and since the IRCEL-CELINE stations are mostly located within source regions, the better correlation of P8 with NO₂ data might be partly fortuitous. Furthermore, as shown in Fig. 11, the NO₂ daytime vertical profiles calculated with

P1 and P8 are very close, implying very similar NO₂ column amounts. The choice of P1 or P8 as PBL option should therefore not have a large impact on WRF-Chem comparisons with APEX or TROPOMI data.

**4.2.2 Role of temporal variability of the emissions**

The temporal variability of the emissions in the model (Sect. 2.2.8) is dominated by a pronounced, sector-dependent diurnal cycle. In addition, relatively weak seasonal and weekly variations are also applied. To highlight the importance of the temporal

variability in the emissions, Fig. 12 compares the results of simulations either ignoring or including this temporal variability.





The constant emissions for the first simulation (MODEL_NoTF) are an average of the temporally varying emissions over the 3-day period. PBL scheme 1 was used in both cases, and corrections for $NO_2$ measurement interferences are taken into account.

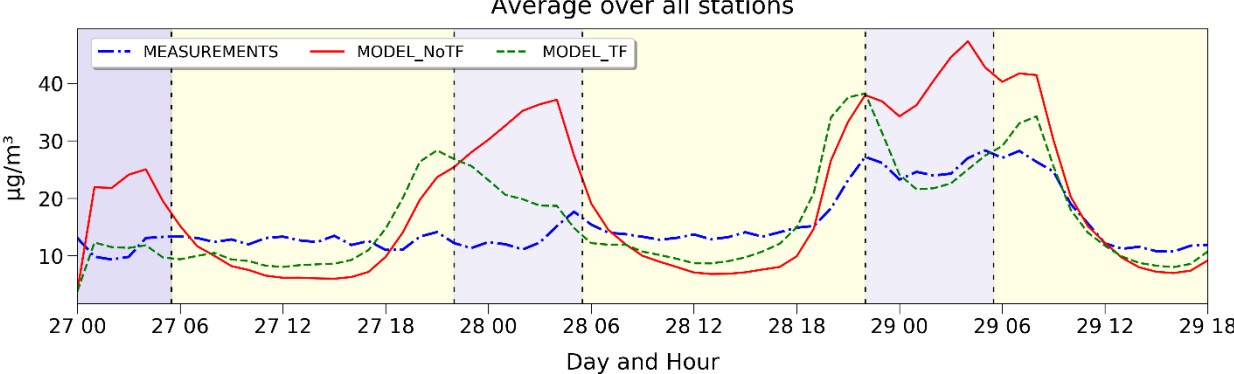

**Figure 12. $NO_2$ modelled and measured concentrations, averaged over all stations. The blue, dash-dotted line represents the observations while the continuous lines represent the model output obtained using either temporally constant emissions (red line) or temporally varying emissions (green line) as described in Sect. 2.2.8. Both model outputs have been corrected to account for measurement interference due to other $NO_y$ reservoirs. The vertical dashed lines separate night- and daytime hours, shown in blue and yellow respectively.**

Although both model results underestimate the daytime $NO_2$ concentrations, applying the temporal factor to the emissions increases those concentrations and improves the agreement with the observations. The effect of adding a temporal profile to the emissions is most evident during the night. The measurements exhibit very little diurnal variation, especially on the 27[th] and 28[th]. When the emissions are constant in the model, the $NO_2$ concentrations are much higher during the night than during the day (by about a factor of 2, see Fig. 12) due to variations in mixing layer height and chemistry. Indeed, lower PBL heights during the night (Sect. 4.1.1) lead to steeper vertical gradients and higher surface concentrations (Fig. 11). In addition, lower levels of the OH radical are expected at night, as its primary sources are photolytic (Logan et al., 1981), leading to longer $NO_2$ lifetimes with respect to loss by reaction with OH. During the nights of the 27[th] and 29[th] of June, these effects are partly or completely offset by the reduced nighttime emissions of the MODEL_TF simulation, leading to a good agreement with the observations. On the 28[th], the nighttime overestimation remains, for reasons unclear.

**4.2.3 Evaluation of the 15-day model simulation against NOx, CO and $O_3$ station data**

Comparison of the 15-day model simulation with measured surface $NO_2$*, NO, CO and $O_3$ data is shown in Fig. 13. The concentrations were averaged over all stations within the inner domain, at which measurements are available for the corresponding species. Both interference-corrected ($NO_2$*) and uncorrected $NO_2$ concentrations are shown in Fig. 13. Simulated $NO_2$* generally follows the diurnal trends seen in the IRCEL-CELINE data, with peaks generally found during the nighttime hours and lower values during the daytime. There is a consistent model underestimation of daytime $NO_2$, although the Lamsal correction intended to account for interferences in $NO_2$ measurements improves the agreement between simulated and observed daytime concentrations. The average model bias for daytime hours (9-17h) is -28% and -40% with and without




the Lamsal correction, respectively. The model overestimation during the nighttime indicates insufficient loss due to vertical mixing and/or chemical sink. For example, heterogeneous conversion of $NO_2$ to HONO and $HNO_3$ on surfaces at the ground or on aerosols (Kleffmann et al., 2003) is not represented in the model. However, this sink should not strongly impact the

nighttime $NO_2$ levels, since the reported conversion rates based on in situ measurements do not exceed ~2% $h^{-1}$ and generally fall below 1% $h^{-1}$ (Kleffmann et al., 2003; Hu et al., 2022). Flaws in the parameterization of vertical transport are suggested by the model underestimation of PBL height during nighttime (Sect. 4.1.1), in particular for the P1 parameterization. Insufficient vertical mixing at night is also suggested by comparison of the observed and measured diurnal cycle of CO concentration, which is presented in the next subsection.

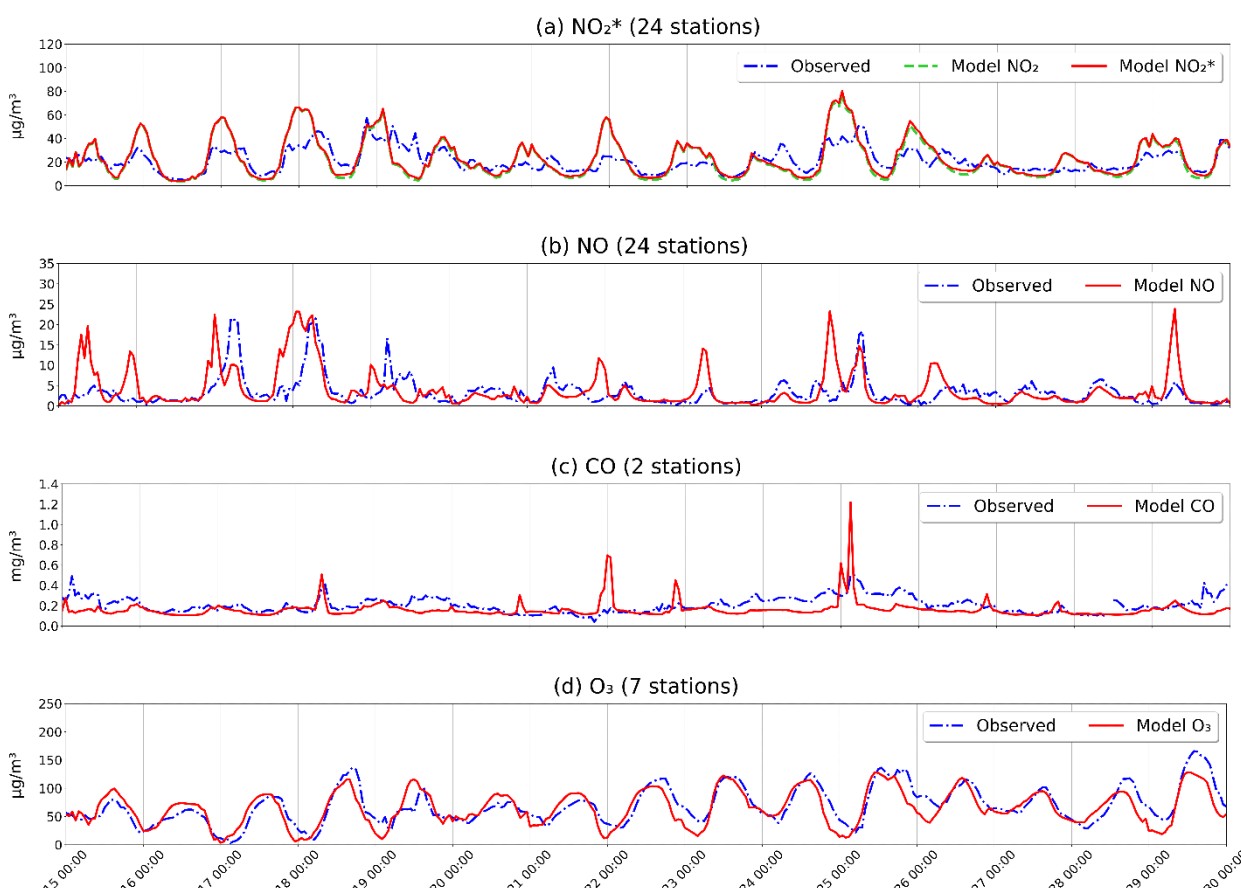


**Figure 13. Time series of observed and modelled concentrations of (a) NO₂\*, (b) NO, (c) CO and (d) O₃ at IRCEL-CELINE network stations for the 15-day duration of the reference simulation. Each time series is an average of measurements at the available stations for the corresponding species. Interference-corrected model NO₂ (NO₂\*) is shown in red, and uncorrected NO₂ in green.**

The $NO/NO_2$ ratio (RNOx) from both the model and observations are compared on Fig. 14. In this instance, the modelled $NO_2$

concentrations are uncorrected, and the measurements are corrected for interferences as described in Sect. 3.2. The modelled ratios match the observations very well, except for a substantial overestimation (factor of 2) on the 29th. The modelled RNOx



closely follows the photochemical steady state (PSS) defined by assuming equilibrium between the $NO_x$ interconversion reactions

$$NO + O_3 \rightarrow NO_2 + O_2 \qquad k_1 \qquad\qquad\qquad\qquad\qquad\qquad\qquad\qquad\qquad\qquad\qquad (R1)$$

$$NO + HO_2 \rightarrow NO_2 + OH \qquad k_2 \qquad\qquad\qquad\qquad\qquad\qquad\qquad\qquad\qquad\qquad (R2)$$

$$NO + RO_2 \rightarrow NO_2 + products \qquad k_3 \qquad\qquad\qquad\qquad\qquad\qquad\qquad\qquad\qquad (R3)$$

$$NO_2 + hv \rightarrow NO + O \qquad J_{NO2} \qquad\qquad\qquad\qquad\qquad\qquad\qquad\qquad\qquad\qquad (R4)$$

NO is chemically converted into $NO_2$ mainly through reactions with $O_3$, $HO_2$ and organic peroxy radicals ($RO_2$), while $NO_2$ undergoes photolysis to produce NO and (upon reaction of atomic oxygen with $O_2$) $O_3$. The rates are obtained from the

chemical mechanism in the model. RNOx at PSS is calculated using

$$\left(\frac{[NO]}{[NO_2]}\right)_{PSS} = \frac{J_{NO2}}{k_1[O_3] + k_2[HO_2] + k_3[RO_2]}, \qquad\qquad\qquad\qquad\qquad\qquad (3)$$

Here the product $k_3[RO_2]$ denotes a sum over all organic peroxy radicals. The time period between the 27th and 29th of June had very little cloudiness (as correctly simulated by the model, see Fig. 7), ensuring low uncertainties in the calculation of the $NO_2$ photolysis rate. The reaction with ozone typically accounts for >95% of the NO-to-$NO_2$ conversion rate, while the sum

of peroxy radical terms makes up the rest. The overestimation of RNOx in the model on the $29^{th}$ is likely partly due to the underestimation of ozone on that day (Fig. 13). In addition, the lower daytime $NO_x$ concentrations on the $29^{th}$, compared to the previous days, bring NO concentrations closer to the detection limit (of the order of 0.5 µg m$^{-3}$) of the chemiluminescence measurement, thereby increasing the observational uncertainties. Another source of uncertainty is the interference in the $NO_2$ measurements, which we corrected following Lamsal et al. (2008). Ignoring this correction would have worsened the model

comparison (due to lower RNOx based on observations). For the period 23-29 June, when excluding the data for which [NO] is very low (i.e. below or equal to 1 µg m$^{-3}$), the average RNOx from the model is 0.323, only 5% higher than in the measurements (0.307), when the correction is applied. The overestimation would be much more significant (26%) without this correction.

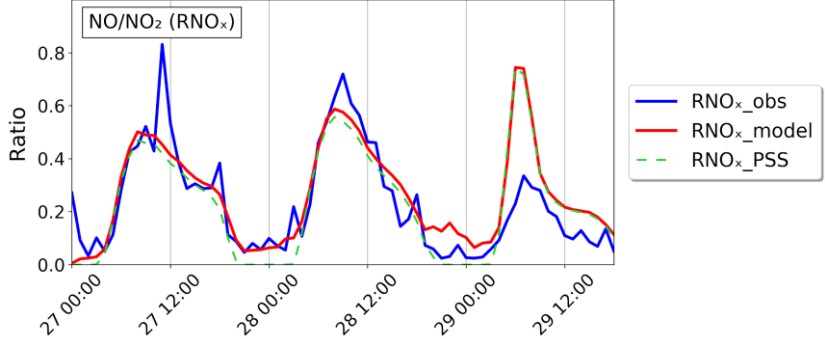




**Figure 14. Observed and modelled (NO/NO₂) ratio (RNOx) on 27-29 June at IRCEL-CELINE sites within the inner domain. Also shown is the ratio at photochemical steady state (RNOx_PSS) based on concentrations and rates from the model run.**

Carbon monoxide, (CO) being a long-lived species, its evolution is mostly flat over the 15 days, with some peaks in concentration that the model is overestimating (22 and 25 June). The concentration of CO shown in Fig. 13 is obtained from only two stations, thus the discrepancy between model and measurements might be due to their limited representativity or to misestimated local emission or transport patterns in the model.

The secondary pollutant $O_3$ is often anti-correlated with its precursor species $NO_x$ (Han et al., 2011). For example, ozone exhibits minimum values at night, when $NO_2$ has its maximum. During the day, photochemistry leads to the formation of ozone and OH radicals, causing the lifetime of $NO_x$ to be minimum. During the night, the suppression of vertical mixing leads to the accumulation of $NO_x$, whereas the downward transport of ozone from higher levels is reduced. The observed daily maximum ozone concentrations (of the order of 100 µg m$^{-3}$ or about 50 ppbv) are usually well reproduced by the model. The nighttime ozone concentrations are frequently underestimated, possibly reflecting insufficient vertical mixing. Nevertheless, the overall good agreement between model and observation indicates that the model is accurate in representing the photochemical processes leading to formation of ozone.

### 4.2.4 Diurnal cycle of surface concentrations

The average diurnal profile of the 4 species is shown in Fig. 15. This distinctly shows the overestimation of both $NO_x$ compounds at night, as well as their underestimation during the day. The $NO_2$ correction for interferences is highest around noon (~30% increase), when the concentration of peroxyacetyl nitrate (PAN) is maximum. PAN is a product of VOC oxidation, which occurs mostly during daytime due to the high levels of OH radicals. The early morning maximum in both $NO_x$ compounds (at or slightly before 6 AM) occurs about one hour before rush hour and the corresponding peak in the emissions (Fig. 4). The afternoon traffic-related emission peak around 4 PM is not visible in the $NO_x$ time series, due to the dominant roles of the chemical sink and boundary layer development. As expected due to its longer lifetime, of the order of 50 days (Müller and Stavrakou, 2005), the diurnal variation of CO is relatively weak. Since the chemical sink is too slow to affect the diurnal shape, and since the CO emissions are maximum during daytime (as for $NO_x$), the nighttime maximum predicted by the model can only be due to the enhanced stability of the PBL. The observed flat diurnal profile suggests that this stability enhancement is exaggerated in the model, in line with the underestimation of modelled PBLH against ceilometer data (Sect. 4.1.1). As discussed above, the daytime buildup of ozone is well reproduced by the model, although there is a slight temporal shift in the peak value, for reasons unclear.



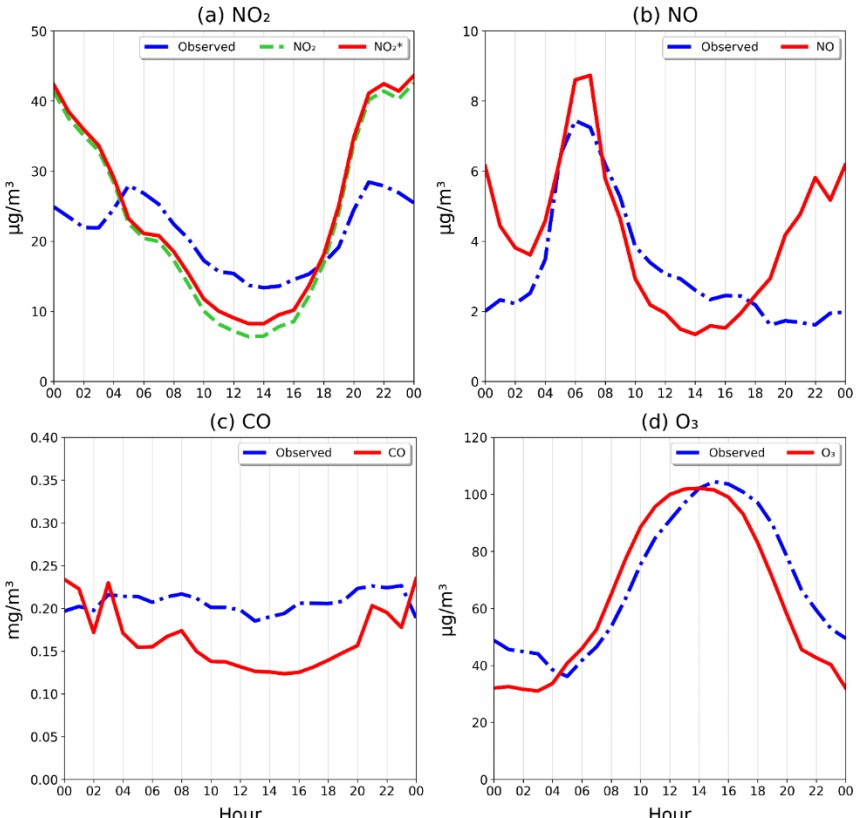

**Figure 15. Average diurnal cycle of measured and modelled concentrations of (a) NO₂\*, (b) NO, (c) CO, and (d) O₃. Model results from reference 15-day simulation (15-29 June). In the case of NO₂, both the interference-corrected and uncorrected model outputs are presented.**

### 4.2.5 Improving the weekly profile of NOₓ emissions based on in situ data

The relative bias (RB) of the model against daytime NO2 data (9-17h) calculated for each day using the (Lamsal-corrected) simulated NO2* is shown on Fig. 16. RB is always negative and systematically higher during weekdays (-37% on average) than during weekends (-10%) and especially the Sundays (-4% on 16 and 23 June). This pattern clearly suggests a misrepresentation of the weekly cycle in the emissions. The weekly shape of emissions suggested by Crippa et al. and adopted in our model simulations shows less variation between weekdays and weekends than expected, based on previous studies (e.g. Stavrakou et al., 2020; Valin et al., 2014). Indeed, both spaceborne NO2 columns (Stavrakou et al., 2020) and in situ NOx data (Valin et al., 2014) were shown to be consistent with emission reductions of the order of 40% during weekends compared to weekdays over U.S. cities. By comparison, the Crippa profiles imply a reduction of only about 15% over the Antwerp area (Fig. 17).





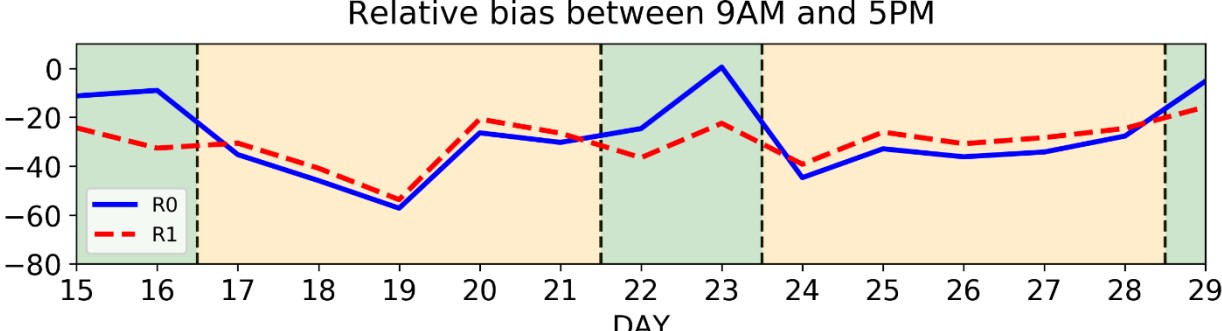

**Figure 16. Average relative bias of the model against NO$_2$ station data, calculated for each day between 9 and 17h, between 15 and 29 June. Blue line: initial run R0; red, dashed line: run R1 with the updated weekly cycle of emissions (see Fig. 17). The vertical dashed lines separate the weekends (green) and weekdays (orange).**

Here we use the IRCEL-CELINE measurements in combination with WRF-Chem simulations to derive a "top-down" emission weekly cycle. Following Zhu et al. (2021), a reference run and a perturbation run with a uniform percentage increase of emissions (+20%) are conducted in order to estimate the sensitivity of NO$_2$ output concentrations to a change in NO$_x$ emissions. In this way, top-down daily emissions are derived, providing an improved agreement with measurements. Those emissions are calculated using

$$\beta = \frac{\Delta E/E_{\text{ref}}}{\Delta C/C_{\text{ref}}}, \tag{4}$$

$$E_{\text{td}} = E_{\text{ref}} \times \left(1 + \beta \frac{C_{\text{obs}} - C_{\text{ref}}}{C_{\text{ref}}}\right), \tag{5}$$

where $E_{ref}$ and $E_{td}$ are the a priori and top-down daily emissions, respectively, $\beta$ is a dimensionless scaling factor reflecting NO$_2$ sensitivity to emission perturbation, $\Delta E$ is the change in emission and $\Delta C$ is the change in output concentration between the reference and perturbation runs. Here the spatial patterns of the emissions are unchanged by the inversion, and both $\beta$ and the scaling factor multiplying the emissions are constant over the model domain. $C_{obs}$ and $C_{ref}$ are averaged concentrations over all stations during daytime (9-17h). Nighttime concentrations are excluded as they are more affected by horizontal and vertical transport variability.

The $\beta$ factor calculated from the two model simulations is close to 1 (in the range 0.95 - 1.15 over the 15 days), i.e. the concentrations are approximately proportional to the emissions. The daily emission scaling factors ($E_{td}/E_{ref}$) for each day of the week are shown in Fig. 17. The emissions are increased by ca. 50% during weekdays, whereas the enhancement is much lower on Saturdays (~20%) and Sundays (~10%). The temporal variation of the emissions needed to match the in situ data is obtained by multiplying the daily-averaged scaling factors by the a priori weekly profile used in the model, i.e. the Crippa profile. Upon normalization, the emission weekly cycle constrained by in situ data is obtained (dashed magenta line on Fig. 17). The normalized emissions during the weekend (ca. 0.7) are in agreement with the study of Stavrakou et al. (2020), exhibiting a lower ratio between weekend and weekday (0.6) than the profile proposed by Crippa (0.87). During the rest of



week, the profile shows unexpected variations, possibly due to model errors in e.g. dynamical fields. Those likely unrealistic variations are removed in the simplified weekly profile derived in this work (black dashed line).

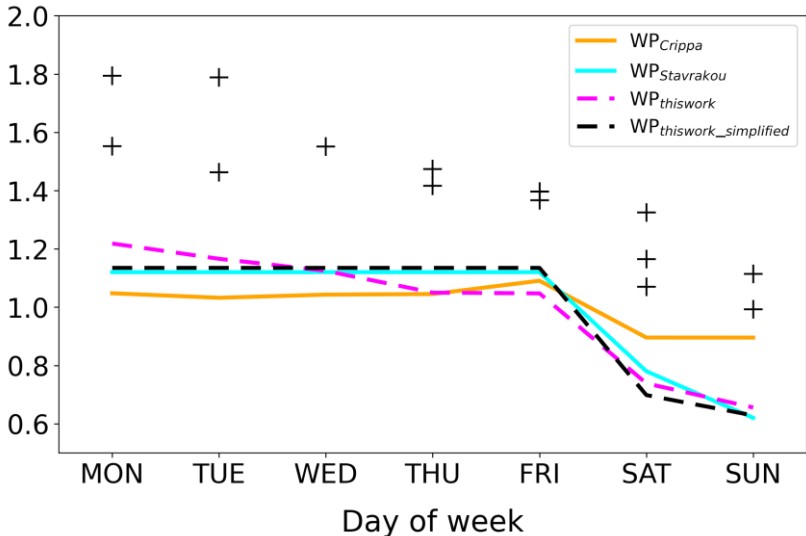


**Figure 17. Normalized weekly profiles (WP) of NO$_x$ emission over the Antwerp area 1) based on Crippa et al. (2020) (orange line), 2) adopted by Stavrakou et al. (2020) (blue line), and 3) derived in this work (dashed magenta) based on the optimization of daily emissions constrained by in situ NO$_2$ data. Also shown is the simplified weekly profile adopted in further model calculations (see text). The crosses represent the daily emission scaling factors needed to match the in situ data in the model. The dashed magenta**
**line is the normalized temporal variation of emissions needed to match NO$_2$ data (obtained by multiplying the daily averaged scaling factors by the weekly profile from Crippa et al. (2020), followed by normalization).**

Note that beside the revised weekly profile, the top-down emissions based on station data imply also a substantial enhancement of the emissions (+43% on average). This enhancement is likely not realistic, as it would lead to large model overestimations against remote-sensing data (APEX and TROPOMI), discussed in the next subsections. The impact of the new weekly profile
is verified through an additional 15-day model simulation (R1) in which the weekly temporal profile from Crippa et al. (2020) is replaced by the new weekly cycle (WP_thiswork_simplified). As seen in Fig. 16, the R1 run generally displays a more constant relative bias of the model against NO$_2$ data over the 15 day time period.

**4.3 Comparison with remote sensing chemical observations**

**4.3.1 Model evaluation against APEX data**

Figure 18 compares the spatial distribution of the WRF-Chem NO$_2$ tropospheric column (from run R0) against the regridded APEX column measurements on the 27[th] and 29[th] of June. Generally, the model set-up is accurate in representing the distribution of NO$_2$ over the two days, with well-defined plumes originating in the industrial areas northeast of the city of Antwerp. The plume orientation, namely southwest and northwest on the 27[th] and 29[th], respectively, is dictated by the wind direction, about 30° on the 27[th] and 120° on the 29[th] (Fig. 7).



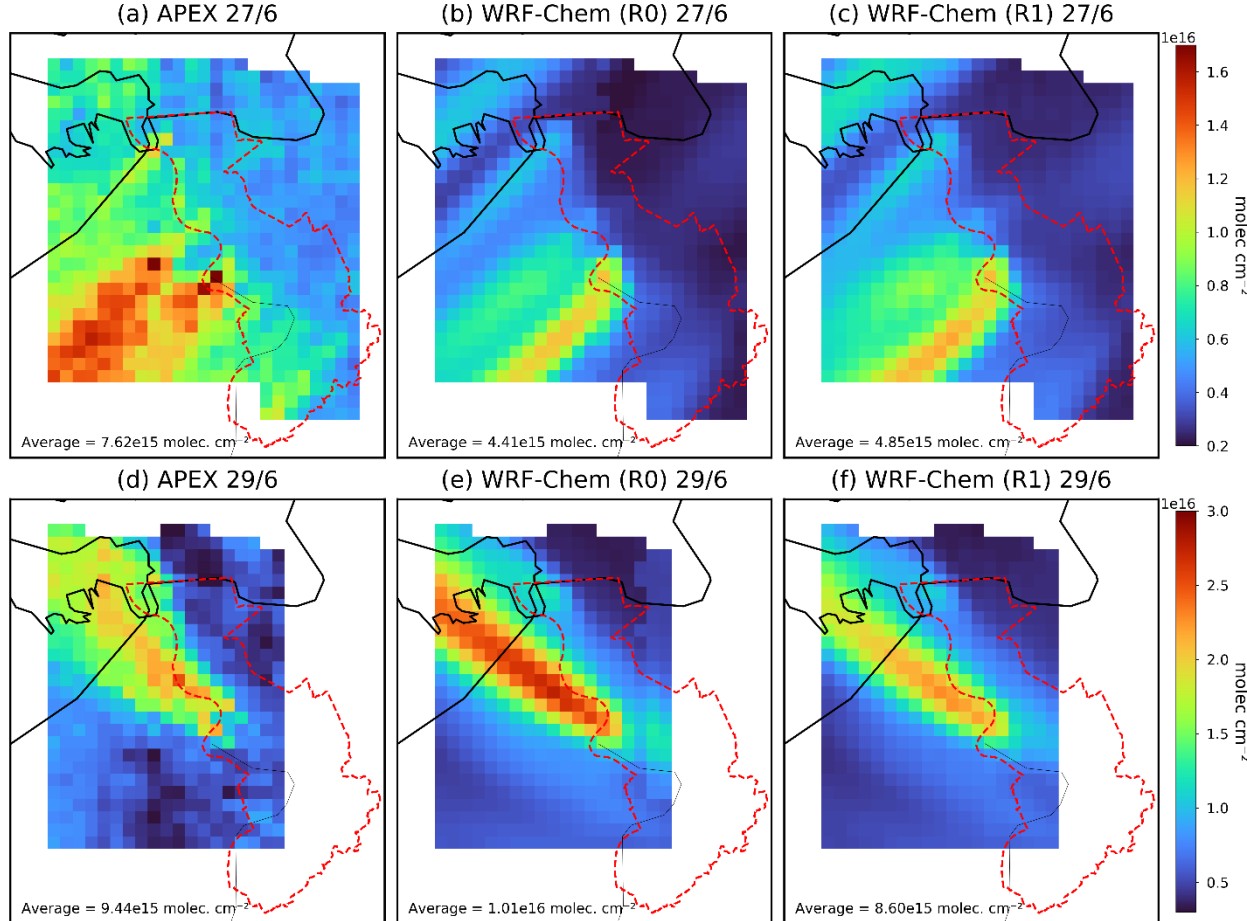

**Figure 18. APEX and corresponding WRF-Chem NO₂ distribution on the 27th (top row) and 29th (bottom row) of June 2019. The APEX data were regridded onto the 1x1 km² model grid. The red dashed outline represents the region of the municipality of Antwerp. Measurements are shown in the leftmost column, while the middle and right rows display the modelled columns obtained from simulations R0 and R1. The average column value is shown within each panel.**

Similarly to the comparison with ground-based measurements, the modelled NO₂ in the reference run (R0) is strongly underestimated on the 27th (weekday, Thursday), while a better agreement is achieved on the 29th, a Saturday. On the 29th, although the regions of low emissions are well represented, the plume is more narrow and too concentrated in the WRF-Chem output, suggesting insufficient dispersion through wind transport and/or overestimation of emission sources along the axis of the plume.

The agreement with APEX data is improved by implementing the new weekly cycle constrained by IRCEL-CELINE data described in the previous section (Fig. 17). The emissions increase on Thursday (by almost 10%) and decrease on Saturday (by ~20%) as result of this change. This improves the consistency in how the model performs over the two days. On the 29th, the mean bias (+7% in R0) becomes negative (-9% in R1). On the 27th, the negative model bias of run R0 (-43% on average) remains large on the 29th (-36%), however. This could be partly due to the larger wind speed overestimation (by ~2 m s⁻¹ on





the 27$^{th}$ vs. ~1 m s$^{-1}$ on the 29$^{th}$, see Fig. 7) leading to excessive evacuation of the pollution plume by horizontal transport, and to enhanced vertical mixing due to wind-favoured turbulence.

### 4.3.2 Evaluation of TROPOMI NO$_2$ based on APEX and WRF-Chem simulation

The modelled NO$_2$ columns are evaluated against near-simultaneous APEX and TROPOMI measurements, whereby the model acts as an intercomparison platform to compare the two measurement techniques and develop on previous validation studies

(Tack et al., 2021) to characterize biases. For a meaningful comparison, APEX data (regridded to the model resolution) and the corresponding model NO$_2$ columns (obtained by convolution of model profiles with the averaging kernels from either APEX or TROPOMI) are regridded to the resolution of TROPOMI. Note that on the 29$^{th}$ of June, two TROPOMI overpasses were available for comparison. Note however that the second overpass (08855) had larger viewing zenith angles in the APEX area (~63°) resulting in about twice larger pixel sizes than those of the 27$^{th}$ and of the first overpass on the 29$^{th}$. Furthermore,

the time difference between the TROPOMI and APEX measurements is larger for the second overpass of the 29$^{th}$. As in Tack et al. (2021), we keep for comparison only data from the first overpass (08854).

As seen in Fig. 19, the model correlates very well with both APEX and TROPOMI data ($R > 0.9$), but it consistently underestimates the observed NO$_2$ columns on the 27$^{th}$ of June, with slopes lower than 1 for both linear regressions. The slopes are higher on the 29$^{th}$, for which an excellent agreement is found between the model and TROPOMI. Interestingly, the ratio of

the two slopes (model vs. APEX and model vs. TROPOMI) is similar for the two days (0.84 and 0.81), suggesting a moderate but consistent underestimation of TROPOMI NO$_2$ with respect to APEX. Note that the TROPOMI underestimation would be more pronounced without application of averaging kernels to the model profiles (slope ratio of ~0.67), as can be seen from the regressions of model columns with APEX and TROPOMI given in the Supplement (Fig. S1).

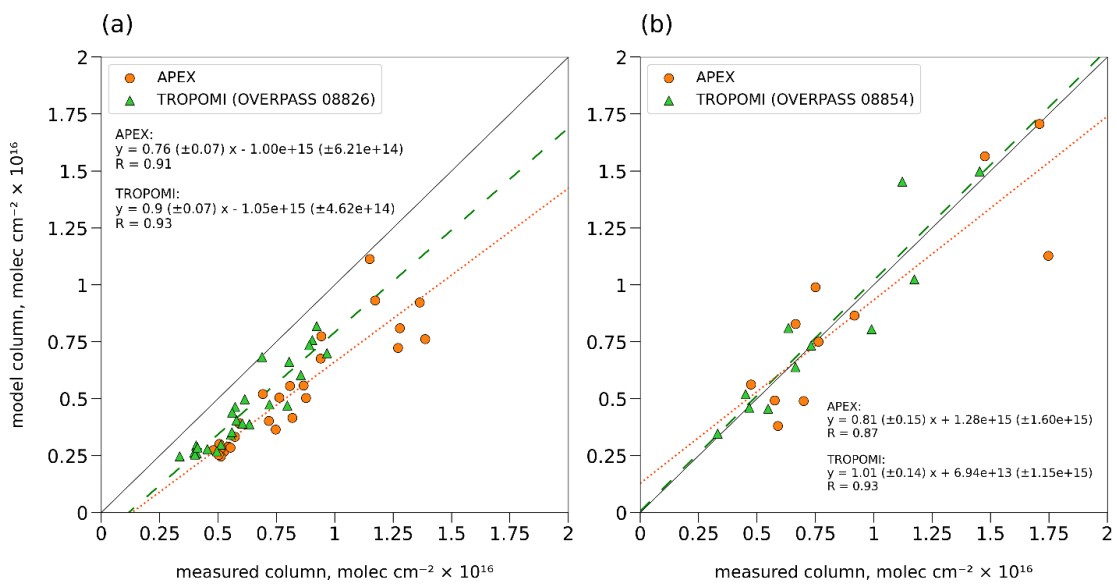

 

**Figure 19. Scatter plots and linear regressions of modelled vs measured columns (APEX and TROPOMI_v1.3) on (a) 27th of June and (b) 29th of June. Orange dots and dotted regression lines are for APEX, green triangles and lines for TROPOMI.**

By combining the linear regressions of the model results against APEX and TROPOMI, a linear relationship between APEX and TROPOMI columns is derived,

$$\text{APEX} = \left(\frac{m_T}{m_A}\right)\text{TROPOMI} + \left(\frac{c_T - c_A}{m_A}\right), \tag{6}$$

where $m_A$ and $m_T$ are the slopes of the linear regressions between the model and APEX and TROPOMI respectively, and $c_A$ and $c_T$ denote their intercepts. Taking the average of the slope and intercept obtained in this way for the two days, and assuming APEX to be the truth, we derive a formula intended to correct for the TROPOMI biases identified above.

This bias correction for the TROPOMI v1.3 product is given by

$$C'_{v1.3} = (1.217 \pm 0.16) \times C_{v1.3} - (0.783 \pm 1.3) \times 10^{15}, \tag{7}$$

where $C'_{v1.3}$ and $C_{v1.3}$ are the bias-corrected and uncorrected TROPOMI columns (molec. cm$^{-2}$), respectively. A similar analysis was performed for the TROPOMI_PAL product (see Fig. S2 in the Supplement), leading to the bias correction formula

$$C'_{PAL} = (1.055 \pm 0.14) \times C_{PAL} - (0.437 \pm 1.3) \times 10^{15}, \tag{8}$$

Note that the bias correction was obtained using TROPOMI data in the approximate range $(4-15) \times 10^{15}$ molec. cm$^{-2}$. The correction might therefore not be applicable outside of this range.

The above regressions were obtained by using the WRF-Chem model in a specific setting, namely, a 15-day simulation adopting scheme P1 as PBL parameterization, with $NO_x$ emissions and temporal variations as described in Sect. 2.2 and Sect. 4.2.5. Since the modelled $NO_2$ vertical profile shapes show some dependence on the choice of PBL scheme (Fig. 11), 1-day sensitivity simulations (starting on 27/6 at 0:00UT) were conducted to estimate the impact of the PBL scheme on the regressions for the 27/6. Only schemes P1, P5, P8, P11 and P12 were tested, since the other schemes (P2, P4, P6, P9 and P10)

were found to deteriorate the model performance against meteorological data and surface $NO_2$ measurements (Sect. 4.1 and 4.2). The regressions of the modelled columns against APEX and TROPOMI show very little dependence on the PBL scheme: for example, the regression slope from the comparison against APEX data varies by less than 1% between the different schemes. The variance of the slopes is only slightly larger (1.5%) for the comparison with TROPOMI, and the variance of their ratio is also very small (1.2%). Similarly, the 15-day run and the 1-day run adopting the same PBL scheme (P1) have

very similar results in comparisons with APEX and TROPOMI. Finally, a 1-day sensitivity run with $NO_x$ anthropogenic emissions enhanced by 43% (achieving a better model agreement with APEX on the 27th as well as with daytime surface $NO_2$ data) increases the slopes of the regressions of the model vs. both APEX and TROPOMI by more than 50%, but leaves their ratio essentially unchanged (+0.5%). These tests show that the above bias-correction formulas are only very weakly dependent on the model settings.



### 4.3.3 Adjustment of emissions using WRF-Chem and TROPOMI data

The TROPOMI and WRF-Chem tropospheric $NO_2$ column distributions are compared on Fig. 20 over the model domain. To minimize the features due to transient transport effects, as well as to reduce the noise, the data was regridded to 0.1° x 0.1° resolution and averaged over the 15 days of the simulation.

Application of the bias-correction to TROPOMI data, as described in Eqn. 7, enhances the columns by up to 10% (or $8x10^{14}$ molec. $cm^{-2}$) over hotspots such as Paris and industrial areas around Antwerp, Rotterdam and the North Rhine/Ruhr region in western Germany. TROPOMI columns below $3.6x10^{15}$ molec. $cm^{-2}$ are decreased by the bias correction, but as discussed in the previous section, the validity of this correction is uncertain for low columns. The model succeeds in reproducing the main hotspots, although it clearly underestimates TROPOMI over Paris, Brussels, Zeebrugge (Belgian coast) and the Northern part of the Ruhr valley. The R1 model overestimates the data over the Rhine valley and Amsterdam. The strongest model overestimation is found in a region to the west of the Rhine, located within the box labeled "PP" on Fig. 20. Several among the largest coal power plants in Germany (Neurath, Niederaussem and Frimmersdorf) are located in this area (http://globalenergyobservatory.org). The strong hotspot north of the city of Antwerp is a special case, with underestimation being found over the harbor of Antwerp and overestimation across the border in Holland. Generally, WRF-Chem R1 underestimates the low $NO_2$ columns, such as over Eastern Netherlands and Flanders, the North Sea, and Northern France. The bias correction improves the model agreement with the observations in these regions, although it brings the TROPOMI columns below the WRF-Chem values over the least polluted areas in Northern France, the Belgian Ardennes and the Eifel plateau.







**Figure 20. Spatial distribution of (a) uncorrected and (b) bias-corrected TROPOMI NO₂ columns (OFFL v1.3.1) alongside WRF-Chem simulations using (c) a priori emissions (R1) and (d) top-down emissions (R2). Model and data were regridded to 0.1° x 0.1° and averaged over the 15-day simulation period. The stippling in panel b indicates pixels for which the bias correction might be invalid (TROPOMI < 4x10¹⁵ molec. cm⁻²). White circles represent cities with >200k population. The dashed white boxes represent four regions of interest: Paris (PA), Brussels-Antwerp (BA), the Ruhr area (RU) and a cluster of power plants in Western Germany (PP).**

Many factors might contribute to the differences between the model and (bias-corrected) TROPOMI, including errors in the model transport and chemistry as well as in the bias correction. However, a major source of error lies in the estimation of the emissions. Here we apply a crude method to correct the spatial distribution of emissions in the model, by making the





assumption that emission errors are the leading reason for the differences with TROPOMI. The method uses equations similar

to those used to amend the weekly cycle (Eqs. 4 and 5), where $C_{ref}$ and $C_{obs}$ denote 15-day averaged WRF-Chem and (bias-corrected) tropospheric $NO_2$ columns, respectively. A reference run and a perturbed run with 20% increased $NO_x$ emissions are used to update the emissions. This emission update is not considered reliable below the validity range of the bias correction, all the more because low $NO_2$ columns are also disproportionately affected by the long-range transport from more polluted areas. By contrast, the $NO_2$ hot spots are primarily due to local emissions. Note however that even the hot spots are affected

by wind transport, which is likely the main source of error in this optimization of emissions. To limit such errors, the emission correction is not kept when the modelled column shows a weak sensitivity to emission changes, i.e. when the β factor (Eq. 4) is significantly higher than unity (more specifically β > 1.45). High values of β (Fig. S3) are found away from the major emission regions and near the borders of the model domain due to the influence of lateral boundary conditions. Over polluted areas, β is generally close to, or even lower than, unity. Lower-than-one values of β indicate that chemical feedbacks amplify

the effect of emission changes on the concentrations. Indeed, in high $NO_x$ areas, increasing $NO_x$ emissions depletes the OH radical through the $NO_2$+OH reaction (Lelieveld et al., 2016), and this reaction is the main sink of $NO_x$. Over low $NO_x$ areas, $NO_2$+OH is negligible as OH radical sink, and $NO_x$ emission increases lead to enhanced $O_3$ and OH levels, mainly due to the $HO_2$+NO reaction which converts $HO_2$ to OH and produces ozone. This explains the higher values of β over more remote areas (such as the Belgian Ardennes), as the $NO_x$ emission increase leads to shorter $NO_x$ lifetimes.


Figure S4 shows the difference between the observed and modelled tropospheric $NO_2$ column for the reference run (R1) and for a run using emissions adjusted as described above (R2). Note that the R1 simulation already performs quite well, since the average model bias over polluted areas (TROPOMI>$4x10^{15}$ molec. $cm^{-2}$) is only -6% in simulation R1 (-4% in run R2). The emission adjustment (Fig. S4(b)) leads to improvements in the match between model and observation, particularly evident in

the North Rhine and Ruhr regions, where both overestimations in the southwestern part (between Cologne and Krefeld) and underestimations in the northern part are now decreased. Emission increases over the Paris and Brussels-Antwerp areas also reduces the overall model underestimation in these regions. To summarize, the negative biases over the Paris area (box labeled PA on Fig. 20), Brussels-Antwerp (BA) and the Ruhr valley (RU) are decreased respectively from -33%, -13% and -14% in simulation R1 to -5%, -6% and -9% in simulation R2 using top-down emissions. These improvements are realized by

increasing the emissions by 39% (PA), 20% (BA) and 13% (RU) on average over each region. At the same time, the model overestimation around the large German power plants (PP) decreases from 19% to 8% thanks to emission decreases averaging -19% in this region. In other areas such as Rotterdam/The Hague, the top-down emissions fail to improve the agreement, however. This is likely due to transport effects since the column in one pixel is dependent on emissions in many neighboring pixels, and the pattern of model biases is particularly heterogeneous around Rotterdam/The Hague and other regions.



**5 Discussion and conclusions**

Several validation campaigns for the TROPOMI $NO_2$ product (version 1.3) have been conducted, mostly in mid-latitude areas in the vicinity of strong emission sources, similar to the region simulated in this study. Methods include comparison with ground-based Differential Optical Absorption Spectroscopy (DOAS) measurements and with airborne spectral imagers (e.g. APEX). Those studies generally report either the relative bias of TROPOMI $NO_2$ with respect to the correlative measurements, or more frequently, the slope ($s$) and intercept ($i$) of regressions of the type $C = i + s\ C'$, where $C$ denotes the TROPOMI column and $C'$ the independent measurements. The relationship derived in this work between APEX and TROPOMI v1.3 (Eq. 7) can be expressed similarly, with $s = 0.82$ and $i = 6.4 \times 10^{14}$ molec. cm$^{-2}$. The relative bias of TROPOMI v1.3 is calculated to be about -10% for columns in the range $(6\text{-}12) \times 10^{15}$ molec. cm$^{-2}$. The TROPOMI_PAL product achieves even lower biases in this range (a few percent), with $s = 0.95$ and $i = 4.1 \times 10^{14}$ molec. cm$^{-2}$.

Based on comparisons with Multi-axis DOAS (MAX-DOAS) measurements at 19 sites worldwide, Verhoelst et al. (2021) inferred negative biases in TROPOMI tropospheric $NO_2$ columns, estimated at -37% in slightly polluted conditions ($2 \times 10^{15}$ molec. cm$^{-2}$) and -51% in highly polluted areas ($12 \times 10^{15}$ molec. cm$^{-2}$). Those biases are much larger than those obtained in this work (about -10%). A first reason for this discrepancy might be the different spatial representativeness of ground-based and spaceborne or airborne data, especially in urbanised/industrial areas. In addition, as pointed out by Verhoelst et al. (2021), large errors in TROPOMI tropospheric columns are due to shortcomings in the a priori $NO_2$ profile used in the TROPOMI product algorithm. Comparison between satellite and independent measurements can be improved by replacing the standard a priori vertical profile of TROPOMI $NO_2$, obtained from the TM5-MP model, with a new profile obtained either from a higher resolution model or from measurements, when available. This highlights the importance of accounting for the difference in a priori profile and vertical sensitivity between TROPOMI and the independent instrument. This was realized in this work (Sect. 4.3.2) by applying averaging kernels from TROPOMI and APEX to the model concentrations when calculating the corresponding model columns. Without the averaging kernel application, the slope $s$ of the relationship between APEX and TROPOMI (calculated from the regressions of Fig. S1) would be 0.67, and the resulting TROPOMI bias would be estimated at -19% for a column of $8 \times 10^{15}$ molec. cm$^{-2}$.

This work uses validation data described and used by Tack et al. (2021) to evaluate TROPOMI. Based on a direct comparison of APEX and TROPOMI, Tack et al. determined TROPOMI $NO_2$ biases of -21% and -15% for the two flights over Antwerp. This is consistent with our comparisons performed when ignoring averaging kernels (Fig. S1). Substitution of the TM5-MP a priori profiles with the high-resolution profiles from CAMS (at 0.1° x 0.1° resolution) in the TROPOMI dataset was found by Tack et al. to improve the comparison considerably, e.g. by lowering the average bias to about -2%. The slope of the regression between TROPOMI and APEX was also improved, from ~0.7 with the original product to 0.83-0.94 with the CAMS-modified dataset. Those values compare well with the slope obtained here (0.82).



Tropospheric NO$_2$ columns were evaluated over the New York area using airborne and ground-based Pandora observations
(Judd et al., 2020). The regression slope of the standard TROPOMI NO$_2$ product against airborne columns was 0.68, which
increased to 0.77 when correcting for the a priori vertical profile of the NO$_2$ product, obtained from the North American
Model–Community Multiscale Air Quality (NAMCMAQ). The slopes for the comparison with ground-based Pandora data
were 0.8 and 0.82 for the TM5 and NAMCMAQ products, respectively, very similar to the slope derived in this study. Similar
results were found by Griffin et al. (2019) using ground-based Pandora column data and airborne concentration measurements
above the Canadian Oil Sands. Biases ranging between -15% and -30% for the original TROPOMI dataset were reduced to
between 0 and -25% when using modified air mass factors based on high-resolution model profiles and improved surface
reflectivity and snow identification.

Pandora measurements at urban and suburban sites in the Greater Toronto Area showed slopes of 0.70-0.77 for the standard
TROPOMI NO$_2$ product, and between 0.76 and 0.85 when updating the a priori NO$_2$ profile to a higher resolution profile shape
and updating the albedo and snow flags (Zhao et al., 2020), also consistent with our findings. In contrast, measurements by
the same group at a rural site (Egbert) indicated overestimations of TROPOMI NO$_2$ relative to the Pandora columns, of about
10-15% for columns of the order of $4 \times 10^{15}$ molec. cm$^{-2}$.

Dimitropoulou et al. (2020) and Chan et al. (2020) evaluated TROPOMI NO$_2$ data in Uccle (near Brussels) and Munich,
respectively, using 2-D MAX-DOAS instruments. Significant negative biases of the original TROPOMI product were found,
amounting to -30% or more at both sites. Upon replacement of the a priori vertical NO$_2$ profiles of the TROPOMI algorithm
with the MAX-DOAS profiles, however, these negative biases are reduced in Munich (to ca. -20%) and disappear almost
completely in Uccle.


To conclude, the moderate TROPOMI v1.3 underestimation (-10%) and slope of regression against APEX data (0.82) obtained
in this study are well in line with previous validation studies in polluted conditions. The agreement between TROPOMI and
correlative tropospheric NO$_2$ measurements shows systematic improvement when the NO$_2$ vertical profile utilized in the air
mass factor calculation of the satellite product is replaced by higher-quality profiles obtained from either measurements (e.g.
MAX-DOAS) or a high-resolution model. Alternatively, the difference in vertical sensitivity and a priori profiles of the two
instruments (when available) can be dealt with through the use of averaging kernels, as in the present study using the WRF-
Chem model as intercomparison platform, or more directly by applying the formalism of Rodgers and Connor (2003), as was
done recently for validating TROPOMI HCHO data using Fourier Transform infrared spectroscopy (FTIR) measurements
(Vigouroux et al., 2020).




Nevertheless, the comparison of spaceborne columns with ground-based optical measurements is made difficult by the high heterogeneity of $NO_2$ abundances, especially near emission hotspots. This is illustrated by the dependence of comparison statistics on spatial coincidence criteria, as seen in many studies (e.g. Zhao et al., 2020; Dimitropoulou et al., 2020; Chan et al., 2020). To a large degree, this representativeness issue disappears when evaluating TROPOMI against APEX data, due to
their very fine resolution (70x120 $m^2$) and large number ensuring high spatial coverage of TROPOMI pixels. Further, adopting a fine-resolution model as an intercomparison platform together with careful model sampling strategy allows to take care of co-location differences as well as differences in vertical sensitivity and a priori profiles. The validation results from this work and from previous studies provide strong evidence that TROPOMI v1.3 $NO_2$ columns are only moderately underestimated under polluted conditions (typically -10% for columns > $4x10^{15}$ molec. $cm^{-2}$) when a priori profile shape differences are
properly accounted for. More work is needed to characterize the performance of TROPOMI in less polluted conditions, although there is evidence of a slight overestimation of low $NO_2$ columns. The overestimation of the high columns essentially disappears in the recently released TROPOMI_PAL product, at least in the column range considered here.

Although the model compares generally well with meteorological observations, it struggles to accurately represent near-surface
wind speed, in line with previous studies indicating wind speed overestimations of the order of 1 m $s^{-1}$ near the surface over Europe (Tuccella et al., 2012). This discrepancy might be partly due to underestimated surface roughness length in WRF-Chem over forests and urban areas, as noted by Shen et al. (2020). This might impact boundary layer mixing as well as horizontal transport processes and adds further uncertainties to comparisons with ground-based chemical observations. The diurnal profile of $NO_2$ is too pronounced in the current model set up, showing too high maxima during the night, and a
consistent underestimation during the day. This could be in part due to a misrepresentation of the diurnal cycle of emissions from Crippa et al. (2020), or to issues with the model transport, including wind speed overestimation and insufficient vertical mixing during the night, leading to a buildup of $NO_2$. In addition, chemistry represents an additional source of uncertainty. The concentrations of OH, the main reaction partner of $NO_x$ during daytime, are strongly impacted by short-lived NMVOC emissions, which could be underestimated. Further work should aim at evaluating those emissions and their chemical
representation in WRF-Chem, for example through comparisons with TROPOMI HCHO column data.

Nevertheless, the model is capable of suitably reproducing the major features of $NO_2$ column distribution over both simulation domains, matching the shape and location of plumes seen from aircraft measurements, and locating hotspots as seen from TROPOMI.


Although inconsistencies between the modelled and observed $NO_2$ columns are partly due to errors in the model transport and chemistry, the distribution of model-data differences allows for evaluation of bottom-up emission inventories. In the regions where high resolution emissions are available, i.e. over Flanders and the Netherlands, the comparison indicates slight overestimations of bottom-up emissions e.g. over Amsterdam, and slight underestimations over Antwerp. Determining which



sectors are responsible for the overestimation will require further investigation. Over the rest of the simulation domain, where EMEP was used as a priori, significant underestimations are found in the region of Paris, where emission increases of about 40% are required to match the observations. The regions of Brussels and Dusseldorf also appear to have localized underestimations in the reference runs. The EMEP emissions from a cluster of power plants in the Rhine region (PP in Fig. 20) appear to be overestimated by more than 20%. Upon further inspection, it appears plausible that the location of the very

large Neurath power plant (51.038 N, 6.611 E) has been misplaced in the EMEP inventory. This would indeed explain the mislocation of the strongest $NO_2$ column hotspot from the model distribution, one 0.1° pixel to the West of Neurath. These results are preliminary, however, and more work will be needed to refine the proposed adjustments to the inventories based on satellite observations. These efforts will eventually help in obtaining more accurate emission estimates, and ultimately provide support to mitigation policies.

**Code availability**

WRF-Chem version 4.1.2 was used alongside WRF Pre-processing System version 4.0, both distributed by NCAR at https://www2.mmm.ucar.edu/wrf/users/download/get_source.html. Accompanying WRF-Chem tools for preprocessing files are provided by NCAR at https://www2.acom.ucar.edu/wrf-chem/wrf-chem-tools-community. Pyproj tool used for emission projections and regridding can be accessed at https://pyproj4.github.io/pyproj/stable/index.html.

Python regridding or column calculation scripts are available upon request.

**Data availability**

Air quality measurements of $NO_2$, NO, CO and $O_3$ were obtained from the IRCEL-CELINE website at https://irceline.be/en/air-quality/measurements/monitoring-stations.

CAM-Chem output files are provided by NCAR and available at https://www.acom.ucar.edu/cam-chem/cam-chem.shtml.

CAMS global reanalysis, provided by the Copernicus Atmosphere Monitoring Service, were accessed at https://ads.atmosphere.copernicus.eu/cdsapp#!/dataset/cams-global-reanalysis-eac4?tab=overview. High resolution emissions for the Netherlands are made available by Netherlands' National Institute for Public Health and the Environment at https://www.emissieregistratie.nl/data/grafieken-en-kaarten.

Global emissions from the Emission Database for Global Atmospheric Research (EDGAR) provided by the European

Commission are available at https://edgar.jrc.ec.europa.eu/emissions_data_and_maps. Gridded European emissions distributed by the EMEP Centre on Emission Inventories and Projections can be found at https://www.ceip.at/the-emep-grid/gridded-emissions. EDGAR temporal profiles described by Crippa et al. (2020) and provided by the European Comission can be obtained at https://edgar.jrc.ec.europa.eu/dataset_temp_profile. TROPOMI S5P operational data can be accessed at https://s5phub.copernicus.eu/dhus/#/home. The reprocessed PAL product is available at https://data-portal.s5p-pal.com/.



Reprocessed emissions and WRF-Chem output files are available upon request.

**Author contribution**

CP conducted the simulations and prepared necessary data, conducted comparisons and drafted the paper. JFM and TS conceptualized the project, supervised the work and aided in interpretation of results. DF helped with computational requirements and advised on the simulations. FT provided APEX measurements and guidance about their usage. NV provided

the VMM emissions inventory. QL provided lidar ceilometer measurements. RVM provided radio- and ozonesonde data. TS, DF, FT, FD, QL and RVM reviewed and gave feedback on the paper. JFM revised and edited the paper.

**Competing interests**

The authors declare that they have no conflict of interest.

**Acknowledgments**

We thank Alex Dewalque for providing surface meteorological data at RMI stations. The work done by the VMM in creating a high-resolution local emission inventory is greatly appreciated.

**Financial Support**

This work was conducted through the support of the Belgian Science Policy Office (Belspo) via the European Space Agency funded PRODEX TROVA-E2 (2020-2023) project.

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
