# Peer review of "Cross-evaluating WRF-Chem v4.1.2, TROPOMI, APEX and in situ NO2 measurements over Antwerp, Belgium"

_EGUsphere, 2022_

## Author Comment (AC1)

**Referee 1:**

**General comments:**

1. Page 6, l.162: 'NCEP GFS': Would it make a difference to try to use the ECMWF products instead to initialize the WRF-Chem model.

   CP: Using ECMWF might improve the meteorology, as this has a higher resolution than the NCEP GFS input files (0.1° compared to 0.25°). As far as we know, the impact of using either dataset as meteorological input has not been investigated for Europe. In any case, our comparisons with meteorological observations indicate a generally good performance of the model for meteorology. An exception might be the vertical mixing in the PBL, which might be underestimated during the night, as discussed in the manuscript. However, this aspect would not be affected by the initial conditions and lateral boundary conditions.

2. The authors provide a detailed overview of the various emissions, with focus on the NOx emissions. However, I miss a reference to a (small and uncertain) soil-NOx emission category. Or is this implicitly included in one of the other categories?

   CP: Yes indeed, soil NO emission are considered in the biogenic emissions category. This is now explicitly stated in subsection 2.2.9.

3. l.225. "About 94% of NOx emissions from Flanders is injected above the surface'. I didn't fully understand this sentence, as I assume that traffic emissions are considered as surface emissions, and consist of a large fraction. Could you possibly clarify/reformulate this statement?

   CP: In fact, even road traffic emissions are emitted above the ground in the VMM inventory, at a height of typically 0.5 m. Of course, those emissions fall within the first model layer. To avoid any confusion, we deleted the first part of the sentence which is not relevant to our study since the first model layer extends up to 50m height.

4. the model-observation discrepancy against surface observations for NO2 is still intriguing. I would find it useful to better understand the reasons for this discrepancy. Could one hypothesis be that the assumed PAN concentrations, required to compute the correction factor 'R', is (significantly) under-estimated? More in general, To what extent can the uncertainty in the correction factor contribute to the discrepancy?

   CP: We thank the referee for the intriguing suggestion. Indeed, the concentrations of PAN could be significantly underestimated if the VOC emissions are too low in the model. Note however that an increase in VOC emissions in the model would increase the sink of $NO_x$ due to PAN and organic nitrate formation, but it would also decrease the $NO_x$ sink due to reaction of $NO_2$ with OH, due to OH consumption by reactions with VOCs. The net effect is therefore difficult to estimate, but it could be significant. As already mentioned in the conclusion section, further work will be needed to assess the potential underestimation of VOC emissions (e.g. through comparisons with TROPOMI HCHO columns) and their effects on $NO_2$. A sentence has been added in the

conclusions of the paper: "In addition, VOC emissions also affect the formation of PAN and organic nitrates, thereby influencing the $NO_x$ sink as well as the interference impacting the $NO_2$ measurement (Eq. (2))."

**Referee 2:**

**General comments:**

The manuscript uses averaging of in situ measurements quite extensively for purposes of evaluation of the model but also as a basis for further calculations. This is understandably convenient in many ways, as e.g. the consolidation of measurements makes presentation more concise and easy to follow, but presents certain challenges as observations are often inhomogeneous and stations not always representative of the entire domain in question.  The authors should discuss this challenges and attempt to justify their averaging strategy and consider adding per station plots in the supplement.

CP: Thank you for the comment. We have amended the relevant section regarding the comparison with ground-based NO2 to include spatial distribution information, mainly through the addition of two plots. The additional text and plots are also shown below:

> *The spatial distribution of the $NO_2$-measuring stations is shown in Fig. 13.*

> *Figure 13 illustrates the spatial distribution of the median (modelled/observed) concentrations among the individual stations. The highest underestimation is found at urban stations, e.g. Borgerhout within the city of Antwerp (time series shown in Fig. S1 in the Supplement), where an underestimation of -75% is seen during daytime hours. A large negative bias (-60%) is also found in the Gent city centre. This might indicate an underestimation of traffic emissions under urban driving conditions or a misrepresentation of transport/mixing in large cities such as street canyon effects (Scaperdas and Covile, 1999). The model underestimation is very low at stations further away from emission sources, as is the case with the Schoten (S on Fig. 13) background station (median ratio = 0.97) (see also Fig. S1). Besides the urban stations, the median ratio ranges between 0.6 and 1.1 during the daytime.*

> *Nighttime surface $NO_2^*$ is overestimated at all stations except Borgerhout, the median ratios ranging between ~1.2 and 1.8.*

[Figure]

*Figure 13. Median (a) daytime and (b) nighttime ratio (model/observation) of $NO_2^*$ concentrations at IRCEL-CELINE stations in the inner domain. A, B, and S denote stations Antwerpen, Borgerhout and Schoten.*

[Figure]

*Figure S1. Time series of observed and modelled concentrations of NO₂* at three IRCEL-CELINE network stations: a background site (Schoten), an industrial site (Antwerpen) and an urban one (Borgerhout). Interference-corrected model NO₂ (NO₂*) is shown in red, and uncorrected NO₂ in green.*

section 2.1.1: Not entirely clear here why this was in two separate runs/periods. Could you please explain in the text?

CP: The simulations were conducted as two separate runs and periods for computational purposes. A sentence was added to clarify this in Section 2.1.1: "For computational reasons, the short runs are preferred for evaluating the different physical parameterizations."

section 2.2.8: The description of the methodology for the calculation of the emissions is quite standard, please consider shortening it.

We shortened the section 2.2.8, but only slightly, as we believe important to describe the temporal variations explicitly.

sections 4.1.2 and 4.1.3: The authors should try to shortly present their motives for comparing various meteorological variables with the model running with different PBL schemes.

CP: As already stated in the beginning of Section 4, the PBL parameterization plays a very important role in the simulation of atmospheric composition. The PBL scheme affects meteorological parameters such as wind speed and direction (as shown in the comparisons), and it has a direct effect on near-surface mixing ratios.

section 4.2.1 Is testing which parameterization(s) work best in a specific case by performing a sensitivity study such as this the indicated way of working with a model like WRF-chem? Or are there other reasons to do this in this work?

CP: Yes, we believe that it is the most meaningful way of working with a model like WRF-Chem. Previous WRF-Chem studies also considered sensitivity tests to evaluate the performance of the model in a specific context. There is no clear guideline telling which parameterization should be used for every situation, which compels users to do their own evaluation.

section 4.2.2 The purpose of this section is not sufficiently explained/supported. The way it is presented, it hardly adds to the analysis. Running the model with temporally constant emissions is not really a sensible option to choose from. The paragraph could be removed, unless the authors make more it transparent how it integrates to the rest of the manuscript.

CP: Thanks for this comment, we agree with the reviewer. The section is now removed.

section 4.3.2: Considering the fact that these bias corrections (formulas 7 & 8) are based on a comparison for two days, over Antwerp and for a certain range of NO2 column values, one is left wondering if/how they could be used in some way outside the frame of this study. The formulas are also mentioned in the discussion later on, but the comparison with other studies there is done by means of of the regression relation between TROPOMI and an independent observation (APEX) and not formulas 7 & 8.

CP: The validity range $((4-15) \times 10^{15}$ molec. cm$^{-2}$) is given in the text. Those values are typical of polluted conditions and therefore the formulas are not valid in rural or remote conditions, as stated in the manuscript. The formulas 7 and 8 are obtained from Equation 6, i.e. they relate APEX and TROPOMI, and therefore they can be compared to other studies, which is precisely what we have done extensively in Section 5. We have shown in much detail that our validation results are well in line with previous validation studies in polluted conditions.

section 4.3.3 Similarly, the authors could comment on the general applicability of the emission adjustments introduced here. Can this crude inversion be proposed as a method that can be used outside this study?

CP: The emission inversion method applied in this work has been used in previous studies, e.g. Vinken et al. (2014), Zhu et al. (2021), Visser et al. (2019). For better results, the process could be applied iteratively, i.e. by re-calculating the β factor and adjustment factor with additional simulations using updated emissions, until convergence (e.g. Wells et al., 2020). This was not done in our study, as the average bias over key polluted areas was found to be already strongly reduced in the first iteration.

*Zhu, Y. et al., Remote Sens., 13, 1798, https://doi.org/10.3390/rs13091798, 2021.*

*Vinken, G. et al., Atmos. Chem. Phys., 14, 10363–10381, https://doi.org/10.5194/acp-14-10363-2014, 2014.*

*Visser, A. et al., Atmos. Chem. phys., 19, 11821-11841, https://doi.org/10.5194/acp-2019-295, 2019.*

*Wells, K. et al., Nature, 585, 225-233, https://doi.org/10.1038/s41586-020-2664-3, 2020.*

**Specific comments:**

l. 14: read "generally good performance". Also, please provide figures to support this qualitative comment.

CP: We corrected "generally". The sentence already provides a quantitative figure for the moderate wind overestimation. For the other meteorological parameters, we prefer to remain qualitative for the sake of brevity, since chemical composition is really the main topic of the study.

l. 28-29: Provide some numbers to support the qualitative remarks.

CP: The sentence has been changed to "The model underestimated both APEX (by ca. -37%) and TROPOMI columns (ca. -25%) on the 27/6,…"

l. 6-12x10^15 molec/cm2. Is this range of values low, high?

CP: This is the range of $NO_2$ columns in the Antwerp area, which is a known hot spot of $NO_2$ columns according to TROPOMI measurements (as mentioned in the manuscript).

l. 71: The official validation reports can also be cited here, found in: https://mpc-vdaf.tropomi.eu/index.php/nitrogen-dioxide?start=7

CP: Done, thanks.

l.75-76 Some references could be added here, e.g.: Ialongo, I., Virta, H., Eskes, H., Hovila, J., and Douros, J.: Comparison of TROPOMI/Sentinel-5 Precursor NO2 observations with ground-based measurements in Helsinki, Atmos. Meas. Tech., 13, 205–218, https://doi.org/10.5194/amt-13-205-2020, 2020. Douros, J., Eskes, H., van Geffen, J., Boersma, K. F., Compernolle, S., Pinardi, G., Blechschmidt, A.-M., Peuch, V.-H., Colette, A., and Veefkind, P.: Comparing Sentinel-5P TROPOMI NO2 column observations with the CAMS-regional air quality ensemble, EGUsphere [preprint], https://doi.org/10.5194/egusphere-2022-365, 2022.

CP: Done.

l. 87: "recent" is a stretch, it is probably not meant in absolute terms but in terms of proximity to the period of the study.

CP: This is the most recent air quality report published specifically by an environmental agency over the region. European reports are more recent, but do not concern Flanders specifically.

l.92-93: Please rephrase, it's probably not a promise anymore.

CP: Done.

l. 136-139: Not clear why the 15-day period was not run as one continuous hindcast and had to be split in partially overlapping smaller runs.

CP: When conducting long simulations, it is advised to split the run into smaller, consecutive runs. This allows the modelled meteorology to not stray too far away from "real" conditions, as represented by the meteorological reanalysis provided by NCEP.

l. 164: Which CAMS model would that be, the global or the regional? Please specify.

CP: CAMS global reanalysis (EAC4) was used in this study. This information was amended in the manuscript.

l.165: What is CAM-chem? Please provide reference.

CAM-chem is the Community Atmosphere Model with Chemistry, which is a component of the Community Earth System Model (CESM) used by NCAR. The output of this model is generally used as the boundary and initial conditions for chemical species in WRF-Chem simulations, as suggested by NCAR. The reference (Lamarque et al., 2012) is now provided.

*Lamarque, J.-F., Emmons, L. K., Hess, P. G., Kinnison, D. E., Tilmes, S., Vitt, F., Heald, C. L., Holland, E. A., Lauritzen, P. H., Neu, J., Orlando, J. J., Rasch, P. J., and Tyndall, G. K.: CAM-chem: description and evaluation of interactive atmospheric chemistry in the Community Earth System Model, Geosci. Model Dev., 5, 369–411, https://doi.org/10.5194/gmd-5-369-2012, 2012.*

l. 212: "the entire model domain": is it d01 or d02?

CP: d01. Text adapted accordingly.

l. 274: "over the two domains": over both domains? What would that mean exactly?

CP: We mean over the large domain. Text adapted accordingly.

l.370 please use standard syntax: "unknown reasons".

CP: Done.

Figure 12: NO2* has been defined already so no real need to explicitly refer to the correction of measurement interference in the caption. The same for various places in the text.

CP: Since section 4.2.2 has now been removed in this version, the caption has also been removed.

l. 569: Please provide reference for the conversion rate.

CP: We calculated the PSS ratio with only $k_1[O_3]$ in the denominator of Eqn. 3, and compared with the full expression. This is where the 95% comes from.

l. 571-573: How relevant could that be? The comparison for NO and NO2 in the afternoon of the 29th appears to be quite good (figure 13).

CP: The comparison is indeed good for $NO_2$ in the afternoon, but not really for NO, which is very low.

l. 607: "unclear reasons"

CP: Done.

l.617: "data" should be specified.

CP: The data used in Valin et al. are OMI $NO_2$ column measurements over the Los Angeles metropolitan area in June 2008. In situ NOx data were actually not used in that study. Text changed accordingly.

l.617: Homogenize "NO2" throughout the manuscript by using subscript.

CP: Done.

l. 804: "a few percent": please provide a figure

CP: Between -2% and +2% in the specified column range. Text adapted accordingly.